# MuRA: Multi-Rank Adaptation for Efficient and Effective Test-Time Vision-Language Generalization

## Abstract

Vision-language models (VLMs) have demonstrated remarkable zero-shot capabilities, but their performance degrades significantly when encountering distribution shifts. Recently, test-time adaptation (TTA) methods have been introduced to enhance VLMs' generalization ability. Among these methods, knowledge-adaptive approaches that incorporate Low-Rank Adaptation (LoRA) into vision models show relatively limited improvement compared to other TTA strategies. Our investigation reveals that the fundamental limitation stems from LoRA's static rank configuration, as visual inputs with varying information densities inherently require different ranks for optimal adaptation. To address this challenge, we propose Multi-Rank Adaptation (MuRA), a dynamic rank selection mechanism that adapts to varying data distributions. MuRA achieves state-of-the-art performance on domain generalization and cross-dataset benchmarks. By restricting adaptation to only the deepest layer, MuRA shortens the gradient backpropagation path, thereby significantly reducing both computational and memory overhead. Our method represents an efficient and effective approach to test-time vision-language generalization. Our code will be released as soon as possible.

## 1 Introduction

Recent large-scale vision-language models (VLMs), such as CLIP (Radford et al., 2021), have demonstrated impressive zero-shot capabilities across a wide range of visual recognition tasks. By training on massive image-text pairs, these models learn aligned representations in a shared embedding space, enabling effective generalization to novel categories and unseen tasks without task-specific fine-tuning. Despite their strong generalization ability, VLMs still suffer significant performance degradation under distribution shifts—arising from domain, style, or acquisition differences. To address this limitation,

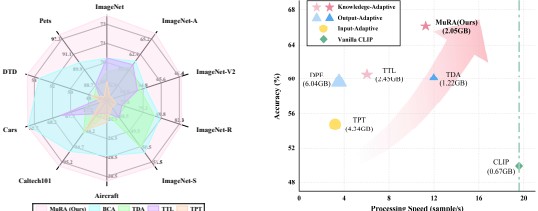

Figure 1: **Effectiveness and efficiency of MuRA.** (**Left**) MuRA consistently surpasses existing methods across diverse datasets. (**Right**) MuRA delivers higher throughput and a reduced memory footprint compared to competing baselines.

recent works have explored test-time adaptation (TTA), a paradigm that adapts pre-trained models to out-of-distribution inputs during inference, using only the test sample without annotation.

As illustrated in Figure 2, existing TTA methods fall into three broad categories: input-adaptive, output-adaptive and knowledge-adaptive approaches. Input-adaptive methods, such as TPT (Shu et al., 2022) and DiffTPT (Feng et al., 2023), optimize learnable prompts or input transformations at test time. Output-adaptive approaches, like TDA (Karmanov et al., 2024), avoid parameter updates entirely and instead recalibrate the output layer using cached feature statistics, enabling efficient adaptation with minimal computational overhead. The third category—knowledge-adaptive methods, such as TTL (Imam et al., 2025), achieve deeper adaptation by injecting lightweight learnable modules (e.g., LoRA (Hu et al., 2022)) and optimizing them during inference. While LoRA-

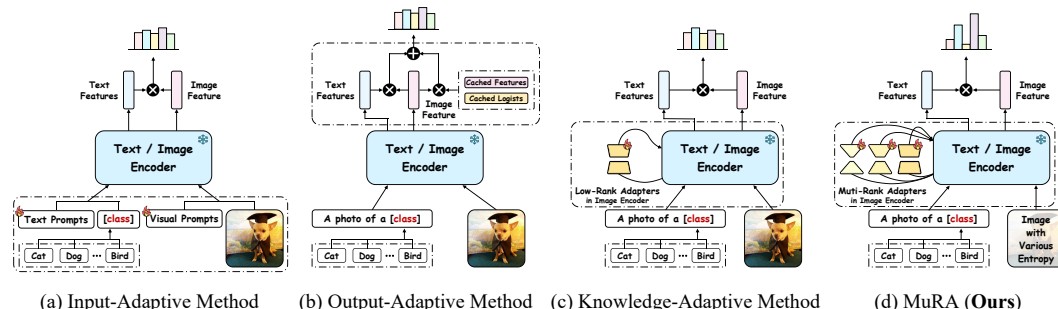

(a) Input-Adaptive Method    (b) Output-Adaptive Method    (c) Knowledge-Adaptive Method    (d) MuRA (**Ours**)

Figure 2: **Overview of test-time adaptation approaches in vision-language models.**

based knowledge adaptation has shown initial promise, its performance gains remain limited compared to other TTA strategies and can even degrade substantially under certain distribution shifts.

Following the paradigm of Parameter-Efficient Fine-Tuning (PEFT), LoRA and its variants (Zhang et al., 2023b; Liu et al., 2024; Hayou et al., 2024) typically rely on careful rank adjustment for optimal performance. Howerver, through systematic evaluation across six diverse vision datasets, we observe that the optimal LoRA rank varies significantly across datasets. As shown in Figure 3 (Left), different domains exhibit distinct rank preferences, revealing the inadequacy of static rank choices. Further analysis of real-world examples suggests that this variation stems from differences in visual complexity. To quantify these distributional differences, we adopt image entropy as a metric for content complexity (see Appendix

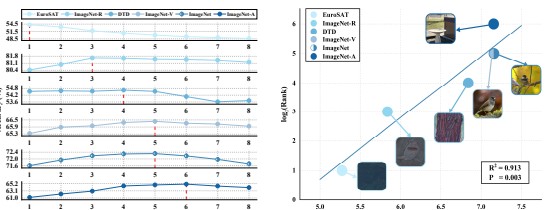

Figure 3: **Analysis of Rank-Performance Relationship and its Correlation with Data Complexity.** (**Left**) Performance of fixed-rank adaptation across six datasets highlights that the optimal rank varies significantly. (**Right**) A strong positive correlation ($R^2 = 0.913$) between the optimal rank ($\log_2$) and data complexity motivates our dynamic rank selection strategy.

A.2 for details). As shown in Figure 3 (Right), image entropy exhibits a strong linear correlation with the optimal LoRA rank ($R^2 = 0.913$): low-entropy images (e.g., textures or stylized art) require only low-rank updates, whereas high-entropy, visually complex scenes (e.g., natural or corrupted images from ImageNet, ImageNet-V) require higher ranks for effective adaptation. This result reveals a fundamental relationship between input complexity and the representational capacity needed for robust adaptation, providing strong motivation for dynamic rank selection in test-time adaptation of vision-language models under distribution shift.

Although recent works (Valipour et al., 2022; Zhang et al., 2023c; Ding et al., 2023) have explored adaptive rank selection in parameter-efficient fine-tuning, they rely on prolonged training with labeled data—rendering them unsuitable for label-free test-time adaptation. To overcome these limitations, we propose Multi-Rank Adaptation (MuRA), a test-time framework that dynamically selects and fuses LoRA modules of varying ranks based on input complexity. MuRA consists of two key components: (1) Multi-Rank Orthogonal Decomposition, which decomposes pretrained weights into rank-specific principal components and residual matrices. The principal components initialize multiple learnable LoRA modules of different ranks. This approach outperforms standard LoRA initialization while mitigating instability issues in input-adaptive TTA methods; (2) Unified Component Fusion, employing a lightweight, learnable router that dynamically computes token-level mixing weights to fuse rank-specific LoRA modules. Unlike conventional TTA methods that reset parameters after each adaptation, we introduce Continuous Router Updating strategy that enables the router to capture evolving feature patterns during adaptation. Through attention visualization, we observe that different ranks capture distinct visual patterns, enabling MuRA to select and combine patterns for more focused visual understanding. In contrast, output-adaptive methods struggle with challenging images due to their reliance on frozen native visual representations.

Extensive experiments on domain generalization and cross-dataset generalization benchmarks demonstrate that MuRA consistently outperforms state-of-the-art TTA approaches. Beyond performance, MuRA is designed to be integrated at the deepest layer of visual models, minimizing gradient propagation path, thereby significantly reducing computational and memory overhead. As illustrated in Figure 1, MuRA achieves both effectiveness and efficiency, making it particularly well-suited for real-world TTA scenarios, where both robustness and speed are critical.

Our main contributions are summarized as follows:

- We identify that optimal LoRA ranks vary significantly across data with different levels of complexity, revealing a critical limitation of fixed-rank adaptation in TTA.
- We propose MuRA, an efficient and effective framework that enables LoRA to dynamically select and integrate appropriate rank components based on visual content, leveraging orthogonal decomposition and router-based fusion during adaptation.
- Extensive experiments demonstrate that MuRA achieves state-of-the-art performance on both domain and cross-dataset generalization benchmarks while maintaining high computational efficiency.

## 2 RELATED WORK

### 2.1 VISION-LANGUAGE MODEL ADAPTATION

Adaptation methods of VLMs mainly fall into prompt tuning-based and feature adapter-based ones. Prompt tuning has emerged as a prominent strategy, particularly effective in few-shot and test-time scenarios. For example, CoOp (Zhou et al., 2022b) optimizes learnable continuous prompt tokens appended to class names, while CoCoOp (Zhou et al., 2022a) introduces a meta-network that generates image-conditioned prompts based on visual features. UPT (Zang et al., 2022) jointly learns both visual and textual prompts through a lightweight network. For feature adapter, it modifies the internal representation of VLMs by inserting lightweight trainable modules. CLIP-Adapter (Gao et al., 2024) adds a residual bottleneck layer to fuse task-specific features, and Tip-Adapter (Zhang et al., 2021) uses a non-parametric cache for similarity-based classification. These methods are parameter-efficient and well-suited to low-data scenarios.

### 2.2 TEST-TIME ADAPTATION WITH VISION-LANGUAGE MODELS

Traditional TTA approaches in computer vision have explored various unsupervised techniques, including entropy minimization, batch normalization calibration, pseudo-labeling, and consistency regularization. In the context of VLMs, several adaptation strategies have emerged. Prompt-based methods like TPT (Shu et al., 2022) optimize text prompts by minimizing marginal entropy across augmented views. DiffTPT (Feng et al., 2023) enhances prompt robustness using diffusion-based augmentations, while PromptAlign (Abdul Samadh et al., 2023) explicitly aligns token distributions between test and source domains. Beyond textual prompts, LoRA-based adaptation has gained traction for its efficiency, with TTL (Imam et al., 2025) adapting CLIP's attention weights during inference. In contrast, training-free methods avoid backpropagation entirely: TDA (Karmanov et al., 2024) refines pseudo-labels via a key-value cache, MTA (Zanella & Ben Ayed, 2024) filters reliable augmentations through inlierness optimization, and DPE (Zhang et al., 2024a) maintains dual semantic and visual prototypes. Recent efficient methods include MCP (Chen et al., 2025) using multi-cache for intra-class compactness, TT-RAA (Fan et al., 2025) leveraging streaming Gaussian databases for retrieval, and GS-Bias (Huang et al., 2025) learning global-spatial biases for logit calibration. BCA (Zhou et al., 2025) introduces an adaptive prior mechanism that updates class embeddings using sample posteriors.

## 3 METHOD

### 3.1 OVERVIEW

As shown in Figure 4, we introduce MuRA, an adaptive framework integrated at the deepest visual layer that learns dynamic rank preferences for optimal adaptation across diverse data distributions. It

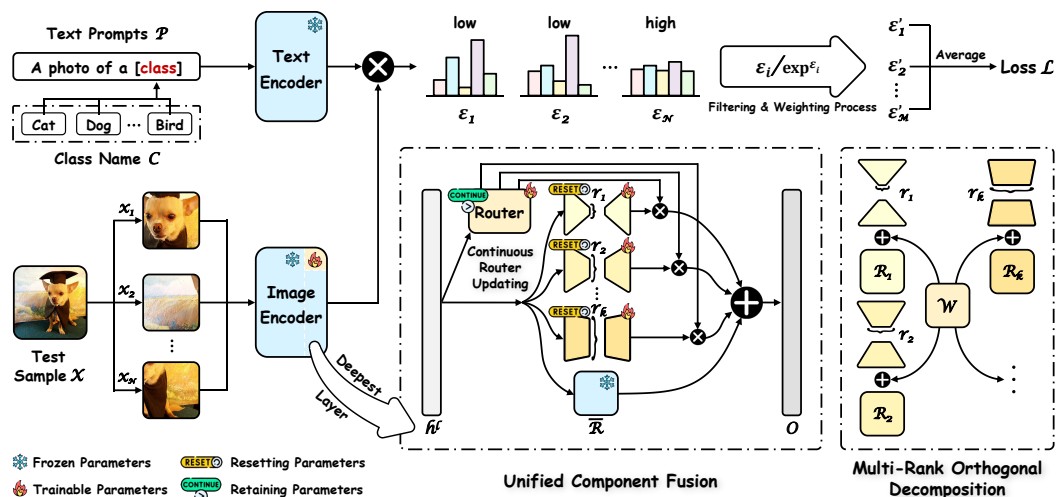

Figure 4: **Overall Architecture of Multi-Rank Adaptation (MuRA).** MuRA is an adaptive framework integrated into the deepest visual layer, designed to overcome the limitations of fixed-rank adaptation. It is built upon two core innovation components: (**1**) **Multi-Rank Orthogonal Decomposition (MROD)**, which initializes rank-specific adaptation components $\{A_i, B_i\}$ and residual matrices $\{R_i\}$ from the pre-trained weight $W$. (**2**) **Unified Component Fusion (UCF)**, which employs a continuously updated Router (CRU) to dynamically combine these different rank components for token-level adaptation. The overall process is optimized via an entropy-based test-time adaptation objective applied to augmented views.

contains two key components: (1) Multi-Rank Orthogonal Decomposition (MROD), which orthogonally decomposes pretrained weights to initialize different-rank LoRA modules, addressing the optimization difficulties caused by zero-initialization in conventional LoRA. (2) Unified Component Fusion (UCF), which combines the matrices decomposed by MROD by first averaging residual matrices $\{R_i\}_{i=1}^k$ into $\bar{R}$, then constructing different low-rank matrices with $\bar{R}$ through a Mixture-of-Experts (MoE) mechanism. The router is designed to update continuously across diverse samples (termed as Continuous Router Updating, CRU), demonstrating effective capture of visual-semantic correlations for token-specific rank selection.

For test-time adaptation on a given input image $x$, we apply a data augmentation function $\mathcal{A}$ to generate a set of augmented views. For each view $\tilde{x} \in \mathcal{A}(x)$, we compute the prediction entropy and apply an adaptive weighting scheme to balance their contributions, following the approach in (Imam et al., 2025). The trainable parameters are updated by minimizing the final loss computed as Eq. 3:

$$\mathcal{H}_\Phi(\tilde{x}) = -\sum_{j=1}^{C} \tilde{p}_\Phi(y_j \mid \tilde{x}) \log \tilde{p}_\Phi(y_j \mid \tilde{x}), \tag{1}$$

$$\beta_\Phi(\tilde{x}) = \frac{1}{\exp(\mathcal{H}_\Phi(\tilde{x}) - \epsilon)}, \tag{2}$$

$$\mathcal{L} = -\frac{1}{\rho N} \sum_{\tilde{x} \in \mathcal{A}(X)} \mathbf{1}[\mathcal{H}_\Phi(\tilde{x}) \leq \tau] \cdot \beta_\Phi(\tilde{x}) \cdot \mathcal{H}_\Phi(\tilde{x}), \tag{3}$$

where $C$ denotes the total number of classes, $\tilde{p}_\Phi$ denotes the predicted class probabilities, $\beta_\Phi(\tilde{x})$ represents the adaptive weighting term as defined in Eq. 2, and $\tau$ is a dynamic threshold set as the $\rho$-percentile of the entropy values across all augmented views.

Upon completion of test-time adaptation, the VLM performs inference on the original image without augmentation. Subsequently, the trainable low-rank matrices are reset to their MROD-initialized values prepared for the next adaptation while preserving the router's accumulated knowledge.

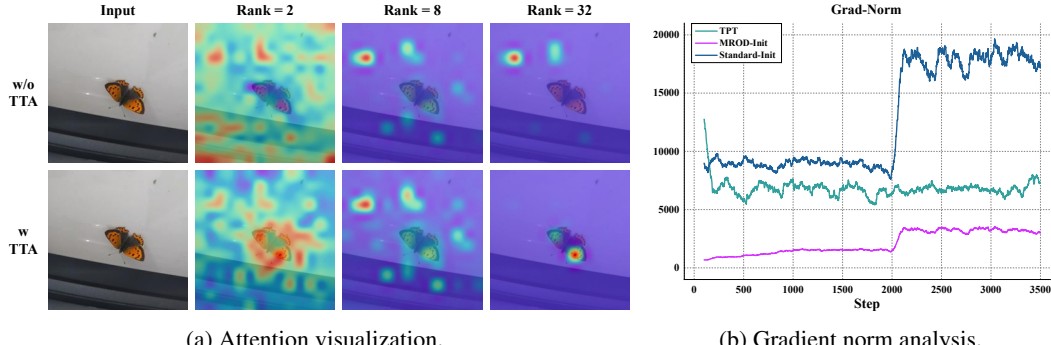

(a) Attention visualization.      (b) Gradient norm analysis.

Figure 5: **The stabilizing effect of MROD-Initialization and Multi-Rank attention refinement.**
(a) Visualizing LoRA attention across ranks. (b) Gradient Norm analysis during adaptation.

## 3.2 Main Components

**Multi-Rank Orthogonal Decomposition.** Following (Meng et al., 2024), we explore initializing
LoRA modules with low-rank principal components obtained through orthogonal decomposition of
pretrained weights. As shown in Figure 5a, different ranks capture distinct and meaningful atten-
tion patterns before adaptation, demonstrating that this pretrained knowledge-based initialization
provides effective knowledge representations for LoRA modules. After TTA, these attention maps
better focus on salient image regions. Moreover, our gradient analysis (Figure 5b) shows that this
initialization achieves smaller and more stable gradient norms than both standard LoRA initializa-
tion and input-adaptive methods like TPT, alleviating the optimization instability during adaptation.

Motivated by these observations, we propose MROD to initialize LoRA parameters through struc-
tured decomposition of pretrained weights. Given rank configurations $\{r_1, ..., r_k\}$ in ascending
order, for a weight matrix $W \in \mathbb{R}^{m \times n}$, MROD performs its economic singular value decompo-
sition: $W = USV^\top$, where $U \in \mathbb{R}^{m \times \min(m,n)}$, $V \in \mathbb{R}^{n \times \min(m,n)}$ are orthogonal matrices, and
$S = \mathrm{diag}(\mathbf{s})$ contains singular values in descending order. For each rank $r_i$, we initialize the LoRA
projection matrices as follows:

$$A_i = U_{[:,:r_i]}S_{[:r_i,:r_i]}^{1/2} \in \mathbb{R}^{m \times r_i}, \tag{4}$$

$$B_i = S_{[:r_i,:r_i]}^{1/2}V_{[:,:r_i]}^\top \in \mathbb{R}^{r_i \times n}, \tag{5}$$

$$R_i = U_{[:,r_i:]}S_{[r_i:,r_i:]}V_{[:,r_i:]}^\top \in \mathbb{R}^{m \times n}, \tag{6}$$

where $R_i$ represents the residual matrix after decomposition. With these initialization strategies, we
can decompose $W$ into:

$$W = A_iB_i^\top + R_i. \tag{7}$$

This orthogonal decomposition ensures $A_iB_i \perp R_i$ for each $i$, facilitating more efficient optimiza-
tion compared to zero-initialization. Note that MROD is performed only once, with these initialized
projection matrices cached for subsequent resets, incurring negligible computational overhead.

**Unified Component Fusion.** To effectively combine different-rank LoRA modules and residual
matrices, we propose UCF, drawing inspiration from Mixture-of-Experts (Jacobs et al., 1991).

The residual matrices from MROD are aggregated through averaging: $\bar{R} = \frac{1}{k}\sum_{i=1}^{k} R_i$. The ag-
gregated matrix $\bar{R}$ replaces the original pretrained weight and remains frozen to maintain pretrained
consistency. Let $h_l$ denote the token representation from the deepest layer. A zero-initialized router
$W_r \in \mathbb{R}^{k \times d}$ computes rank preferences through: $\boldsymbol{\pi}(h_l) = \mathrm{softmax}(W_rh_l)$, where $d$ denotes the
feature dimension. The router initialization ensures stable initialization by producing uniform soft-
max outputs. The final feature output $o$ is obtained through:

$$o = \sum_{i=1}^{k} \pi_i(h_l)A_iB_i^\top h_l + \bar{R}h_l. \tag{8}$$

Table 1: Performance comparison on ImageNet and its OOD variants. Average denotes the mean accuracy across all datasets, while OOD Average excludes ImageNet. The best results are in **bold**.

| Category | Method | ImageNet | ImageNet-A | ImageNet-V2 | ImageNet-R | ImageNet-S | Average | OOD Average |
|---|---|---|---|---|---|---|---|---|
| w/o TTA | CLIP-ViT-B/16 | 68.34 | 49.89 | 61.88 | 77.65 | 48.24 | 61.20 | 59.42 |
| Input Adaptive | TPT | 68.98 | 54.77 | 63.45 | 77.06 | 47.94 | 62.44 | 60.81 |
| | DiffTPT | 70.30 | 55.68 | 65.10 | 75.00 | 46.80 | 62.28 | 60.52 |
| | C-TPT | 69.30 | 52.90 | 63.40 | 78.00 | 48.50 | 62.40 | 60.70 |
| | PromptAlign | - | 59.37 | 65.29 | 79.33 | 50.23 | - | 63.55 |
| Output Adaptive | TDA | 69.51 | 60.11 | 64.67 | 80.24 | 50.54 | 65.01 | 63.89 |
| | MTA | 70.08 | 58.06 | 64.24 | 78.33 | 49.61 | 64.06 | 62.56 |
| | DPE | 71.91 | 59.63 | 65.44 | 80.40 | **52.26** | 65.93 | 64.43 |
| | BCA | 70.22 | 61.14 | 64.90 | 80.72 | 50.87 | 65.37 | 64.16 |
| Knowledge Adaptive | TTL | 70.23 | 60.51 | 64.55 | 77.54 | 48.61 | 64.29 | 62.80 |
| | **MuRA (Ours)** | **72.46** | **66.15** | **66.48** | **82.47** | 51.76 | **67.86** | **66.72** |

Distinct from LoRA parameters, which store specific semantic knowledge and are susceptible to error accumulation from continuous updating, the router learns a robust "Complexity-Capacity Mapping." Accordingly, we propose Continuous Router Updating (CRU) that maintains the router's state throughout adaptation (see Algorithm 1 for implementation details), which empirically demonstrates effective capture of fine-grained visual-semantic patterns for rank selection.

## 4 EXPERIMENTS

### 4.1 EXPERIMENTAL SETTINGS

**Benchmarks.** We evaluate MuRA on two distinct benchmarks: the out-of-distribution (OOD) benchmark and the cross-domain benchmark. The OOD benchmark assesses model robustness on four ImageNet (Deng et al., 2009) variants: ImageNet-A (Hendrycks et al., 2021b), ImageNet-V2 (Recht et al., 2019), ImageNet-R (Hendrycks et al., 2021a), and ImageNet-S (Wang et al., 2019). The cross-domain benchmark evaluates adaptation capability across ten diverse domains: Aircraft (Maji et al., 2013), Caltech101 (Fei-Fei et al., 2004), Cars (Krause et al., 2013), DTD (Cimpoi et al., 2014), EuroSAT (Helber et al., 2019), Flower102 (Nilsback & Zisserman, 2008), Food101 (Bossard et al., 2014), Pets (Parkhi et al., 2012), SUN397 (Xiao et al., 2010), and UCF101 (Soomro et al., 2012).

**Implementation Details.** We employ CLIP with ViT-B/16 backbone as the visual encoder. The rank configuration is set to {2,4,8,16,32}. For each test image, we generate 63 augmented views. We optimize the parameters for a single step using AdamW (Loshchilov & Hutter, 2017) optimizer, with learning rates of 6e-3 and 1e-4 respectively for principal components and router.

**Compared Methods.** We compare MuRA against three categories of approaches: (1) Input-adaptive methods that optimize learnable prompts during inference, including TPT, DiffTPT, C-TPT (Yoon et al., 2024), and PromptAlign; (2) Output-adaptive methods that recalibrate predictions using caches, including DPE, TDA, BCA (Zhou et al., 2025), and MTA (Zanella & Ben Ayed, 2024); (3) Knowledge-adaptive methods that update lightweight modules within CLIP, including TTL and our MuRA. We also include vanilla CLIP as the baseline.

### 4.2 PERFORMANCE COMPARISON

**Results on OOD Benchmark.** As shown in Table 1, MuRA achieves strong performance across ImageNet variants. Notably, on ImageNet-A, which contains naturally adversarial examples, MuRA achieves 66.15% accuracy, surpassing both input-adaptive methods and output-adaptive methods by a large margin. For the challenging ImageNet-R with significant style shifts, our method reaches 82.47%, demonstrating stronger robustness than the static-rank approach TTL (77.54%). While DPE achieves the best performance on ImageNet-S, MuRA still obtains competitive results. Overall, MuRA achieves the best average performance (67.86%) and OOD average (66.72%), validating our

Table 2: Performance comparison on the Cross-Domain benchmark. Average denotes the mean accuracy across all ten datasets. The best results are highlighted in **bold**.

| Category | Method | Aircraft | Caltech101 | Cars | DTD | EuroSAT | Flower102 | Food101 | Pets | SUN397 | UCF101 | Average |
|---|---|---|---|---|---|---|---|---|---|---|---|---|
| w/o TTA | CLIP-ViT-B/16 | 23.67 | 93.35 | 65.48 | 44.27 | 42.01 | 67.44 | 83.65 | 88.25 | 62.59 | 65.13 | 63.58 |
| Input Adaptive | TPT | 24.78 | 94.16 | 66.87 | 47.75 | 42.44 | 68.98 | 84.67 | 87.79 | 65.50 | 68.04 | 65.10 |
| | DiffTPT | 25.60 | 92.49 | 67.67 | 47.65 | 43.13 | 70.15 | **87.23** | 88.22 | 65.74 | 62.67 | 65.17 |
| | C-TPT | 23.50 | 94.10 | 66.70 | 46.80 | 46.70 | 69.90 | 84.30 | 87.40 | 66.60 | 66.70 | 65.30 |
| | PromptAlign | 24.80 | 94.01 | 68.50 | 47.24 | 47.86 | 72.39 | 86.65 | 90.76 | 67.54 | 69.47 | 66.92 |
| Output Adaptive | TDA | 23.91 | 94.24 | 67.28 | 47.40 | 58.00 | 71.42 | 86.14 | 88.63 | 67.62 | **70.68** | 67.53 |
| | MTA | 25.20 | 94.21 | 68.47 | 45.90 | 45.36 | 68.06 | 85.00 | 88.24 | 66.67 | 68.69 | 65.58 |
| | DPE | 28.95 | 94.81 | 67.31 | 54.20 | 55.79 | 75.07 | 86.17 | 91.14 | **70.07** | 70.44 | 69.40 |
| | BCA | 28.29 | 94.85 | 68.86 | 53.49 | **58.89** | **75.12** | 85.97 | 90.45 | 68.41 | 67.95 | 68.23 |
| Knowledge Adaptive | TTL | 23.82 | 93.63 | 67.97 | 46.69 | 42.02 | 70.48 | 85.05 | 88.72 | 66.32 | 69.20 | 65.39 |
| | **MuRA (Ours)** | **30.96** | **95.33** | **68.88** | **54.79** | 57.05 | 74.54 | 85.57 | **92.61** | 69.18 | 70.31 | **69.92** |

motivation that adaptive rank selection better handles varying data complexities. We further evaluate MuRA against other methods on ViT-L/14, with detailed results presented in Section C.1.

**Results on Cross-Domain Benchmark.** Table 2 shows MuRA's superior generalization ability across diverse domains. Compared to DPE and BCA which excel in specific domains, MuRA demonstrates more balanced performance across different visual characteristics. Notably, on the visually ambiguous Aircraft dataset and texture-discriminative DTD dataset, MuRA's multi-rank adaptation mechanism achieves superior performance of 30.96% and 54.79% respectively, substantially outperforming existing methods. The strong overall performance (69.92% average accuracy) across all domains further demonstrates the effectiveness of our approach.

## 4.3 ABLATION STUDY

**Design Analysis.** To validate the effectiveness of our proposed MuRA framework, we conduct comprehensive ablation studies across three challenging OOD datasets: ImageNet-A, ImageNet-R and UCF101. Table 3 presents a systematic investigation of our main designs. The first row represents VLM without any adaptation. The second row shows multi-rank adaptation with standard LoRA initialization. The third row shows single-rank LoRA with rank 62 using MROD-initialization. The fourth row extends to multi-rank scenarios by incorporating UCF to dynamically combine MROD-decomposed modules. The last row further enhances

Table 3: Design ablation study of our MuRA.

| Main Designs | | | Img-A | Img-R | UCF | Avg. |
|---|---|---|---|---|---|---|
| MROD | UCF | CRU | | | | |
| × | × | × | 49.89 | 77.65 | 65.13 | 64.22 |
| × | ✓ | × | 60.61 | 81.26 | 69.15 | 70.34 |
| ✓ | × | × | 63.93 | 81.63 | 69.23 | 71.60 |
| ✓ | ✓ | × | 64.24 | 81.85 | 69.47 | 71.85 |
| ✓ | ✓ | ✓ | **66.15** | **82.47** | **70.31** | **72.98** |

UCF with the CRU strategy. The empirical results demonstrate consistent improvements across all proposed designs. Specifically, MROD-initialization brings significant improvements even in single-rank setting, achieving 7.38% average accuracy gain over the vanilla CLIP. Furthermore, the CRU strategy effectively leverages accumulated knowledge for better rank routing, leading to an additional 1.13% improvement in average accuracy over the model with MROD and UCF only.

**Composition of Ranks.** In Figure 6a, we investigate the effect of rank configurations across three representative datasets. The performance curves show significant improvements as ranks progressively increase up to {2,4,8,16,32}, beyond which the gains plateau while computational costs continue to rise. Therefore, we choose {2,4,8,16,32} as the optimal configuration that balances effectiveness and efficiency. The substantial gap between our progressive strategy and homogeneous configurations validates that adapting to varying data complexities requires a spectrum of rank components rather than a uniform rank setting.

**MuRA Location.** Figure 6b and Figure 6c demonstrate two key findings that validate our design choices. First, progressively incorporating more attention components (Q, K, V, O) leads to consistent performance improvements, supporting the full attention adaptation strategy. Second, adapting the deepest layer significantly outperforms bottom-layer adaptation across all datasets. This deepest-

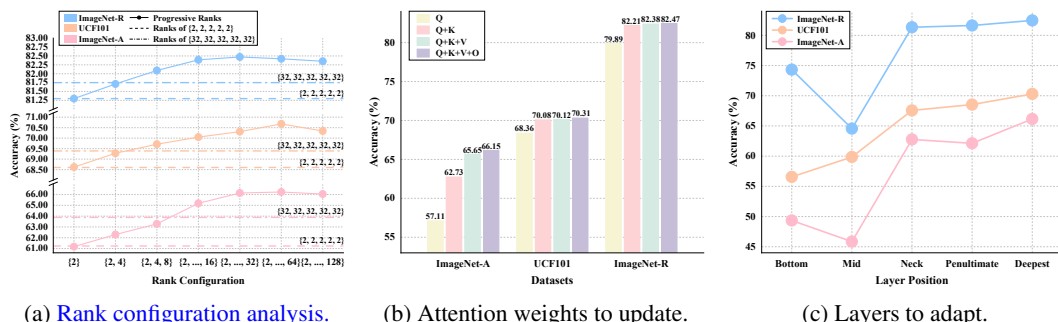

| (a) Rank configuration analysis. | (b) Attention weights to update. | (c) Layers to adapt. |

Figure 6: Analysis of (a) different rank configurations across ImageNet-A, ImageNet-R, and UCF101 datasets, where solid lines represent rank configuration while dashed lines indicate homogeneous low-rank {2,2,2,2,2} and high-rank {32,32,32,32,32} settings, (b) attention weight adaptation, and (c) layer selection (bottom: layers 1-4, mid: layers 5-8, neck: layers 9-10, pendultimate: layer 11, deepest: layer 12).

Table 4: Ablation study of different router designs. mAcc represents the mean accuracy.

| Router Design | mAcc | | |
|---|---|---|---|
| | w/o CRU | w CRU | $\Delta$ (%) |
| Instance-Level | 71.86 | 71.89 | +0.03 |
| **Token-Level** | 71.85 | **72.98** | **+1.13** |

Table 5: Ablation of different routing strategies on various datasets. A: ImageNet-A, R: ImageNet-R, UCF: UCF101.

| Routing Strategy | A | R | UCF | Avg. |
|---|---|---|---|---|
| Hard Routing | 63.17 | 81.66 | 69.89 | 71.57 |
| **Soft Routing** | **66.15** | **82.47** | **70.31** | **72.98** |

layer design not only proves effective but also minimizes the gradient backpropagation path, thereby reducing memory and computational overhead, making MuRA both effective and efficient.

**Instance-level vs Token-level Routers.** We compare two router architectures: Instance-Level router that assigns uniform rank preferences for an entire image, and Token-Level router that enables independent rank selection for each feature token. As shown in Table 4, without CRU, both routers achieve comparable performance (71.86% vs. 71.85% mAcc), as router reinitialization limits their learning of rank preferences. However, when CRU is enabled, the Token-Level router shows significant gains (+1.13% mAcc), whereas the Instance-Level shows negligible gain. This demonstrates that effective rank adaptation requires token-level granularity—coarse, image-wide decisions are insufficient to capture local variations in visual complexity. Moreover, fine-grained routing unlocks the full potential of the CRU strategy, enabling more adaptive and context-aware feature modulation.

**Hard vs Soft Routing.** We compare two routing strategies: Soft Routing that combines different rank by their computed preferences, and Hard Routing that only activates the rank with highest preference score. Table 5 shows that Soft Routing consistently outperforms Hard Routing across all datasets, with particularly notable gains on ImageNet-A (+2.98%). This empirically validates that combining multiple rank components is more effective than selecting a single rank, as it allows for more flexible adaptation to varying feature complexities.

**Accuracy-Speed Analysis.** As illustrated in Figure 1, our proposed MuRA achieves superior performance while maintaining high computational efficiency. Specifically, MuRA achieves 66.2% accuracy with only 2.05GB memory consumption and processes 8.5 samples per second, establishing a new state-of-the-art efficiency-performance balance.

## 4.4 VISUALIZATION

**Attention Analysis.** To qualitatively assess MuRA, we visualize the self-attention heatmaps of CLIP, MuRA, and the individual rank components within MuRA, along with their top-3 prediction probabilities. As shown in Figure 7, different rank-specific LoRA components focus on distinct types of image information: low-rank matrices tend to produce diffuse attention over background

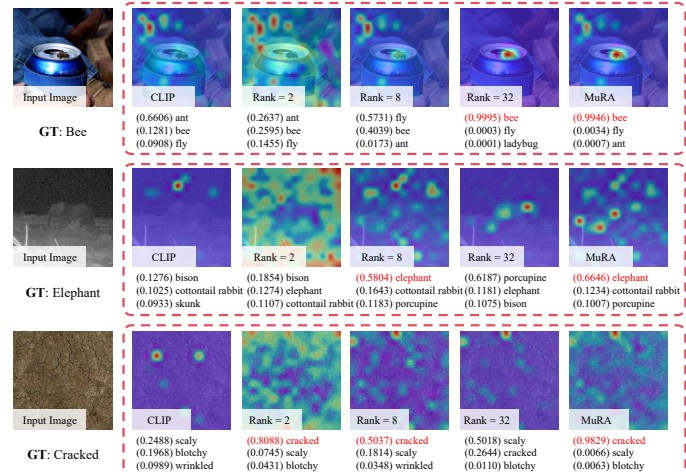

Figure 7: Visualization of attention maps and top-3 predictions for CLIP, different rank components (Rank-2, 8, 32), and MuRA.

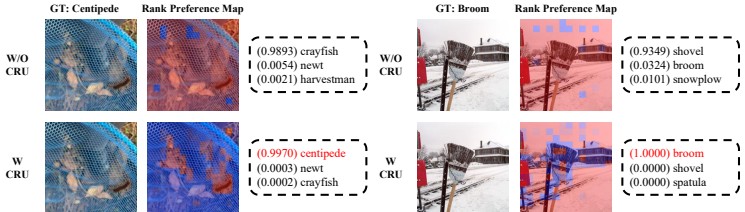

Figure 8: Visualization of rank preference maps and top-3 predictions with and without CRU strategy.

or coarse features, while high-rank ones concentrate on semantically rich foreground areas. Importantly, MuRA dynamically selects appropriate rank components for accurate visual understanding. For object-centric samples like the bee (Row 1), it leverages higher-rank components (Rank-32) for semantic-rich foreground attention. In contrast, for texture-centric samples like the cracked earth (Row 3), it utilizes lower-rank components (Rank-2) to capture broader attention distributions and necessary low-level texture features. CLIP's attention appears to be confined to limited semantic regions, possibly due to its contrastive pretraining nature, which may limit the effectiveness of output-adaptive methods that rely on frozen VLM representations.

**Rank Preference Analysis.** To understand how the CRU strategy influences MuRA's rank selection, we visualize the rank preference maps in Figure 8. In these maps, warmer colors (red) indicate higher-rank component selection, and cooler colors (blue) represent lower-rank preferences. Result shows that CRU enables more semantically meaningful rank allocation, leading to finer-grained rank adaptation. This adaptive behavior allows MuRA to better comprehend unseen test images by dynamically adjusting rank preferences based on local image semantics.

## 5 CONCLUSION

In this paper, we identified that fixed-rank configurations in existing test-time adaptation methods fail to handle varying data complexities. To address this, we proposed MuRA, which leverages multi-rank experts for dynamic capacity matching. We employ MROD for diverse expert initialization and UCF for effective rank aggregation, alongside CRU to refine shared routing knowledge. Extensive experiments demonstrate that MuRA achieves state-of-the-art performance while maintaining high efficiency attributed to our deepest-layer design, validating the superiority of our proposed approach.

## REPRODUCIBILITY

We include all necessary details to facilitate the reproducibility of our work. The experimental setup, including benchmarks, model configurations, hyperparameters, and evaluation protocols, is thoroughly explained in the experiments section. We also give an algorithm in the Appendix to provide the detailed process of our method. We will make our code publicly available.

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

## A  ADDITIONAL DETAILS ON PRELIMINARIES

### A.1  CLIP ARCHITECTURE

CLIP (Radford et al., 2021) consists of image encoder $Ev(\cdot)$ and text encoder $Et(\cdot)$ that map inputs into a shared embedding space. For zero-shot classification, the prediction probability for a test sample $X$ is computed as follows:

$$p(y_i \mid X) = \frac{\exp\left(\mathrm{sim}(\boldsymbol{t}_i \cdot \boldsymbol{v})/\tau\right)}{\sum_{i=1}^{K} \exp\left(\mathrm{sim}(\boldsymbol{t}_i \cdot \boldsymbol{v})/\tau\right)} \tag{9}$$

where $\mathrm{sim}(\cdot)$ denotes cosine similarity, $\boldsymbol{v}$ represents the visual feature vector of the input sample extracted by $Ev(\cdot)$, $\boldsymbol{t}_i$ represents the text feature vector corresponding to the $i$-th class extracted by $Et(\cdot)$. And $\tau$ is the temperature parameter, $K$ is the total number of classes.

### A.2  IMAGE ENTROPY

To quantitatively assess image complexity, we adopt image entropy (Wu et al., 2013) as our measurement metric. Image entropy characterizes the information content of an image, making it suitable for analyzing complexity variations across datasets. The entropy $H$ for a given image is calculated across all color channels as:

$$H = -\sum_{c \in \mathcal{C}} \sum_{i=0}^{L-1} p_c(i) \log_2 p_c(i) \tag{10}$$

where $\mathcal{C} = \{R, G, B\}$ represents the RGB channels, $p_c(i) = \frac{h_c(i)}{M \times N}$ denotes the probability distribution of intensity value $i$ in channel $c$. Here, $h_c(i)$ is the frequency count, $L = 256$ represents the intensity range, and $M \times N$ is the total number of pixels.

### A.3  LOW-RANK ADAPTATION

Low-Rank Adaptation (LoRA (Hu et al., 2021)) is designed to efficiently adapt large pre-trained models by introducing trainable rank decomposition matrices. For a pre-trained weight matrix $W \in \mathbb{R}^{d \times k}$, LoRA parameterizes its update as follows:

$$\hat{W} = W + \Delta W = W + AB^\top \tag{11}$$

where $A \in \mathbb{R}^{d \times r}$ and $B \in \mathbb{R}^{k \times r}$ are trainable low-rank matrices, with rank $r \ll \min(d, k)$. During adaptation, only $A$ and $B$ are updated while the original weights $W$ remain frozen. This significantly reduces the number of trainable parameters from $d \times k$ to $r(d + k)$.

## B  MORE METHOD DETAILS

Algorithm 1 presents the pseudocode of MuRA, summarizing the test-time adaptation pipeline.

The implementation of MuRA will be provided in the supplementary material as a zip file. We also plan to publicly release the full codebase in the near future to facilitate reproducibility and to support further research in adaptive vision-language modeling.

## C  ADDITIONAL EXPERIMENTAL ANALYSIS

### C.1  PERFORMANCE WITH LARGER MODEL.

We evaluate MuRA on a larger vision-language model, ViT-L/14, and compare its performance against the vanilla CLIP and state-of-the-art TTA methods, including TDA and DPE. As shown in Table 6, MuRA achieves an average accuracy of 78.06%, significantly outperforming all baselines. Notably, on ImageNet-A—a benchmark featuring natural adversarial examples and representing the most challenging distribution shifts—MuRA attains a remarkable accuracy of 81.96%.This corresponds to improvements of +28.08% over CLIP, +20.69% over TDA, and +20.87% over DPE. These

---

**Algorithm 1** Multi-Rank Adaptation (MuRA)

---

**Input:** Pre-trained CLIP model $f$, unlabeled test samples $\{x_n\}_{n=1}^N$, ranks $\{r_1 < \cdots < r_k\}$, augmentation set $\mathcal{A}$.

**Definitions:** Optimizer $O$, total parameters $\Phi$, clean optimizer state $S_{\text{empty}}$, router indices $\mathcal{I}_{\text{router}}$.

 1: **Initialization:**
 2: Decompose weights in $f$ using MROD: $W = A_i B_i^\top + R_i$ for each $r_i$;
 3: Initialize router weights $W_r \leftarrow \mathbf{0}$;
 4: Set trainable parameters $\Phi \leftarrow \{(A_i, B_i)_{i=1}^k, W_r\}$;
 5: Initialize Optimizer $O$ on $\Phi$;
 6: $S_{\text{empty}} \leftarrow \text{DeepCopy}(O.\text{state\_dict}())$;     {Save clean optimizer state}
 7: $\mathcal{I}_{\text{router}} \leftarrow$ Identify indices of parameters in $W_r$;
 8: **for** $n = 1$ to $N$ **do**
 9: ***Step 1: Continuous Router Updating (CRU)***
10: Keep router $W_r$ updating;
11: Reset $(A_i, B_i)_{i=1}^k$ to initial values
12: $S_{\text{curr}} \leftarrow O.\text{state\_dict}()$;
13: $H_{\text{router}} \leftarrow \{S_{\text{curr}}[\text{'state'}][j] \mid j \in \mathcal{I}_{\text{router}}\}$;  {Extract Router momentum}
14: $S_{\text{new}} \leftarrow \text{DeepCopy}(S_{\text{empty}})$;    {Reset Adapters to clean state}
15: $S_{\text{new}}[\text{'state'}].\text{update}(H_{\text{router}})$;     {Inject Router momentum}
16: $O.\text{load\_state\_dict}(S_{\text{new}})$;
17: ***Step 2: Test-Time Adaptation***
18: Generate augmented views $\tilde{x} \in \mathcal{A}(x_n)$;
19: Compute prediction probabilities $\tilde{p}_\Phi(y|\tilde{x})$;
20: Calculate entropy $\mathcal{H}_\Phi(\tilde{x})$ and adaptive weights $\beta_\Phi(\tilde{x})$;
21: Compute Loss $\mathcal{L}$ based on entropy minimization;
22: $O.\text{zero\_grad}()$; $\mathcal{L}.\text{backward}()$; $O.\text{step}()$;      {Update $\Phi$}
23: ***Step 3: Inference***
24: Disable gradient computation;
25: Output prediction $y_n \leftarrow f(x_n; \Phi)$; .
26: **end for**

---

results highlight MuRA's strong scalability to larger models and its ability to effectively unlock their adaptation potential.

Table 6: Performance comparison of different methods using larger backbone on various distribution shifts. The best results are in **bold**.

| Method | ImageNet | ImageNet-A | ImageNet-V2 | ImageNet-R | ImageNet-S | Avg | OOD Avg. |
|---|---|---|---|---|---|---|---|
| CLIP-ViT-L/14 | 74.04 | 53.88 | 67.69 | 87.42 | 63.18 | 69.31 | 68.13 |
| TDA | 76.28 | 61.27 | 68.42 | 88.41 | 64.67 | 71.81 | 70.69 |
| DPE | 77.87 | 61.09 | 70.83 | 89.18 | **66.33** | 73.06 | 71.86 |
| TTL | 75.62 | 65.21 | 69.43 | 87.46 | 63.51 | 72.25 | 71.40 |
| **MuRA (Ours)** | **78.66** | **81.96** | **73.19** | **91.79** | 64.69 | **78.06** | **77.91** |

## C.2 ABLATION STUDY ON LOSS DESIGN

Following (Imam et al., 2025), we adopt an adaptive weighting strategy that assigns higher weights to uncertain views in the loss function. We evaluate its effectiveness through ablation studies.

As shown in Table 7, incorporating adaptive weighting brings consistent improvements across all datasets, yielding an average accuracy gain of 0.26% (72.98% vs. 72.72%). The improvement is universal, with the most notable gain on ImageNet-A (+0.43%). These results validate the entropy-based adaptive weighting effectively balances the influence of diverse augmented views, ensuring that informative but uncertain samples are not dominated by high-confidence ones during test-time adaptation.

Table 7: Ablation study on the effectiveness of adaptive weighting mechanism. Results show consistent improvements across all datasets.

| Loss Design | ImageNet-A | ImageNet-R | UCF101 | Avg. |
|---|---|---|---|---|
| w/o Adaptive Weighting | 65.72 | 82.25 | 70.18 | 72.72 |
| **w/ Adaptive Weighting** | **66.15** | **82.47** | **70.31** | **72.98** |

## C.3 ANALYSIS OF ADAPTATION STEPS

We analyze the impact of the number of test-time adaptation steps on model performance. As shown in Table 8, a single adaptation step achieves the best overall accuracy (72.98%), surpassing multi-step variants with 2 steps (72.68%) and 4 steps (72.56%). This indicates that MuRA can effectively align the model to distribution shifts within a single update, while additional steps lead to marginal performance degradation—likely due to over-adaptation on limited or noisy test samples. Moreover, the single-step adaptation significantly reduces computational overhead during inference, avoiding repeated forward-backward passes and maintaining low latency.

Table 8: Performance comparison (%) with different test-time adaptation steps. Interestingly, single-step adaptation achieves the best overall performance.

| Steps | ImageNet-A | ImageNet-R | UCF101 | Avg. |
|---|---|---|---|---|
| **1** | 66.15 | **82.47** | **70.31** | **72.98** |
| 2 | **66.37** | 82.28 | 69.39 | 72.68 |
| 4 | 66.03 | 82.18 | 69.47 | 72.56 |

## C.4 COMPARISON WITH RECENT EFFICIENT METHODS

To further demonstrate the superiority of MuRA, we compare it against the most recent test-time adaptation approaches, including MCP (Chen et al., 2025), GS-Bias (Huang et al., 2025), and TT-RAA (Fan et al., 2025).

As presented in Table 9, MuRA explicitly outperforms these state-of-the-art methods. Specifically, compared to the strong competitor MCP++, MuRA achieves higher accuracy on the most challenging benchmarks (e.g., +6.37% on ImageNet-A and +0.71% on ImageNet-V2). Furthermore, MuRA surpasses TT-RAA by a large margin on ImageNet-A (+5.56%) and ImageNet-R (+1.89%). While GS-Bias focuses on logit calibration, MuRA's multi-rank adaptation proves more effective in handling severe distribution shifts, leading to a substantial improvement in OOD Average accuracy (+6.03% compared to GS-Bias). These results reaffirm that MuRA establishes a new state-of-the-art for efficient test-time adaptation.

Table 9: Performance comparison with recent efficient test-time adaptation methods on ImageNet and its OOD variants. The best results are in **bold**.

| Method | ImageNet | ImageNet-A | ImageNet-V2 | ImageNet-R | ImageNet-S | OOD Avg. | Average |
|---|---|---|---|---|---|---|---|
| MCP | 72.37 | 59.71 | 65.16 | 81.59 | 53.89 | 65.09 | 66.54 |
| MCP++ | **72.64** | 59.78 | 65.77 | 81.73 | **54.39** | 65.42 | 66.86 |
| GS-Bias | 69.02 | 54.55 | 63.37 | 76.64 | 48.21 | 60.69 | 62.36 |
| GS-Bias + E. | 70.57 | 56.61 | 64.62 | 80.49 | 50.33 | 63.01 | 64.52 |
| TT-RAA | — | 60.59 | 64.69 | 80.58 | 49.98 | 63.96 | — |
| **MuRA (Ours)** | 72.46 | **66.15** | **66.48** | **82.47** | 51.76 | **66.72** | **67.86** |

## C.5 SUPERIOR SCALING PROPERTY OF MuRA

To validate the superior scaling capacity of MuRA and isolate the mechanism's contribution, we first align the parameter budget of the TTL baseline to match MuRA's default configuration. As shown

in Table 10, MuRA significantly outperforms TTL ($\sim 7.64\%$ on ImageNet-A), substantiating that MuRA's efficacy is attributable to the dynamic multi-rank design, not passive parameter inflation.

To further illustrate the divergent scaling behavior, we visualize the performance across all configurations (three MuRA points and two TTL points) in Figure 9, where Accuracy represents the average performance across ImageNet-A, ImageNet-R and UCF101 datasets. This visualization reveals a fundamental divergence: MuRA exhibits clear positive scaling (a consistent upward trajectory) with increasing parameters, confirming the multi-rank design's efficient exploitation of additional capacity. In stark contrast, TTL demonstrates severely constrained scaling, showing a performance drop as parameters are augmented. This indicates that its static rank choice fundamentally restricts its adaptability, serving as a critical bottleneck that cannot be overcome by simply increasing the parameter count, as motivated by the findings in Figure 3. In summary, MuRA's multi-rank design provides both superior performance at equivalent capacity and robust scaling ability, a key advantage over fixed-rank methodologies.

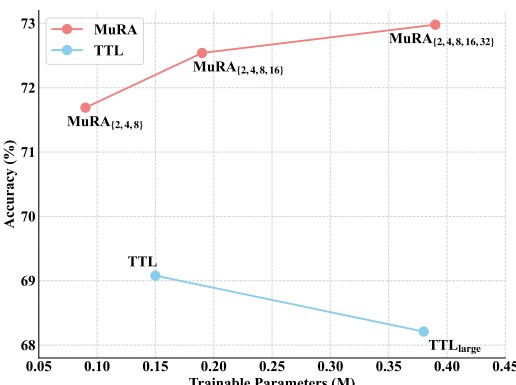

Figure 9: Visualization of the scaling capacity comparison.

Table 10: Performance Comparison under various trainable parameter budgets. MuRA maintains significant superiority even when TTL is scaled up to a comparable parameter count (0.38M). The term '(default)' refers to the original configuration reported in the respective method's publication.

| Method | Trainable Params | ImageNet-A | ImageNet-R | UCF101 | Avg. |
|---|---|---|---|---|---|
| **MuRA$_{\{2,4,8,16,32\}}$ (default)** | 0.39M | **66.15** | **82.47** | **70.31** | **72.98** |
| MuRA$_{\{2,4,8,16\}}$ | 0.19M | 65.19 | 82.39 | 70.05 | 72.54 |
| MuRA$_{\{2,4,8\}}$ | 0.09M | 63.28 | 82.09 | 69.71 | 71.69 |
| **TTL$_{large}$** | 0.38M | 58.51 | 77.60 | 68.52 | 68.21 |
| TTL (default) | 0.15M | 60.51 | 77.54 | 69.20 | 69.08 |

## C.6  SENSITIVITY ANALYSIS OF CONTEXTUAL ROUTER UPDATE FREQUENCY

A key advantage of MuRA's multi-rank architecture is its ability to sustain and accumulate contextual routing knowledge over time via the Continuous Router Updating (CRU) mechanism. To validate the benefit of this continuous accumulation, we conduct a sensitivity analysis by introducing a forced reset frequency to the router state, as visualized in Figure 10. The highest average accuracy ($\sim 73.0\%$) is achieved at our proposed setting (Reset Times = 0), where the router state continuously retains prior knowledge. As the reset frequency increases (Reset Times > 0), performance gradually declines, indicating that the degradation is due to the loss of accumulated knowledge. Crucially, the performance converges toward the baseline strategy without CRU (conceptually Reset Times $\to \infty$), which represents

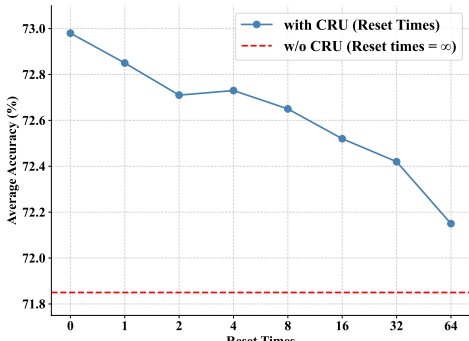

Figure 10: Sensitivity of MuRA to router reset times.

the state of complete loss of prior information. This demonstrates that continuous knowledge allows the dynamic rank selection to gradually route subsequent inputs to more appropriate ranks, leading to improved performance, whereas frequent resetting introduces unnecessary volatility and destroys valuable context history.

## C.7 ROBUSTNESS AND EFFICIENCY UNDER LIMITED AUGMENTATION VIEWS

To highlight MuRA's excellent performance-efficiency balance, we compare MuRA against advanced adaptive methods under varying numbers of input augmentation views (from 8 to 64).

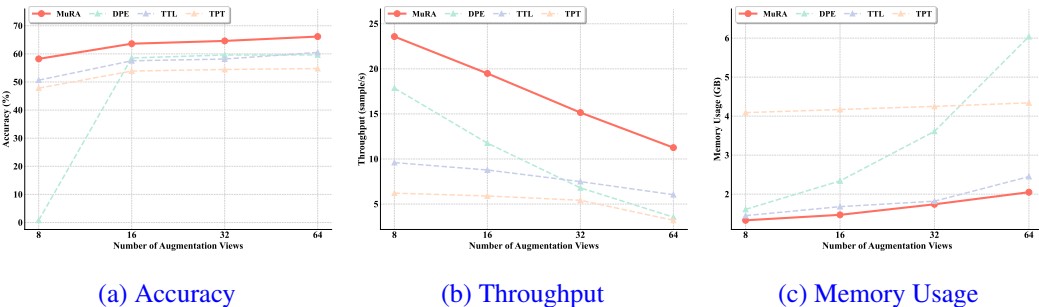

(a) Accuracy    (b) Throughput    (c) Memory Usage

Figure 11: **Robustness and Efficiency under Limited Augmentation Views.** Comparison of accuracy, throughput, and memory usage across varying numbers of augmentation views. MuRA exhibits superior robustness and maintains the best performance-efficiency balance across all view settings.

As shown in Figure 11(a), MuRA exhibits superior robustness: performance is minimally affected by reducing the number of views and remains the highest across all settings. This is a direct consequence of MuRA's core design—it dynamically selects the optimal adaptive capacity (rank) based on the instance-wise input, minimizing reliance on gathering reliable statistics from multiple external views. In contrast, output-adaptive methods like DPE suffer severe performance collapse, especially at extreme limited settings (e.g., 8 views), as they lack sufficient data to fully comprehend and adapt to the new data. The analysis of computational efficiency in Figure 11(b) and Figure 11(c) further highlights MuRA's advantage. MuRA maintains the highest throughput (processing speed) and minimal memory usage across all view settings. This exceptional efficiency originates from MuRA's design, which localizes the multi-rank parameters exclusively to the final layer, maximally reducing the computational and memory overhead. Consequently, MuRA achieves the best performance-efficiency balance among all compared adaptive methodologies.

## C.8 SOURCE OF EFFICIENCY

We investigate the source of MuRA's efficiency by analyzing the relationship between adaptation depth and model performance. Deep-layer adaptation is inherently more efficient as it minimizes the gradient backpropagation path, reducing memory usage and computational cost. However, a critical challenge prevents existing methods from fully exploiting this efficiency.

As demonstrated in Table 11, single rank methods suffer from performance degradation when restricted to the deepest layers. Their performance peaks at the shallow "Bottom" layers (67.87%) and deteriorates to its lowest accuracy at the "Deepest" layer (67.51%), creating a trade-off between efficiency and accuracy.

In contrast, MuRA mitigates this limitation (as also visualized in Figure 6c). Our method exhibits a reverse trend, where performance improves as adaptation moves to deeper layers, peaking at 72.98% at Layer 12. This unique characteristic allows MuRA to utilize the deepest-layer adaptation to maximize throughput and minimize memory footprint without the performance penalty observed in single-rank approaches, effectively unlocking the efficiency benefits of deep-layer adaptation.

## C.9 EFFECTIVENESS OF MROD STRATEGY

Our proposed **Multi-Rank Orthogonal Decomposition (MROD)** fundamentally employs an SVD-based approach to decompose pre-trained weights. As demonstrated in the main paper (see Figure 5), this approach provides superior initial knowledge representation (Figure 5a) and optimization stability (Figure 5b) compared to standard initialization methods.

Table 11: Performance comparison (Average Accuracy across ImageNet-A, ImageNet-R, and UCF101) of Single-Rank vs. Multi-Rank (MuRA) method across different layers. The layer groups correspond to: Bottom (Layers 1-4), Mid (Layers 5-8), Neck (Layers 9-10), Penultimate (Layer 11), and Deepest (Layer 12).

| Method | Bottom | Mid | Neck | Penultimate | Deepest |
|---|---|---|---|---|---|
| Single-Rank | **67.87** | 67.46 | 67.71 | 67.78 | 67.51 |
| **MuRA (Ours)** | 60.08 | 56.76 | 70.56 | 70.77 | **72.98** |

To empirically validate the choice of MROD, we conducted a controlled comparison against two alternative construction strategies. To ensure a fair comparison, all methods share an identical parameter budget and initialize the backbone by aggregating rank-specific residuals:

- **Random Construction:** Utilizes Kaiming uniform initialization for the low-rank matrices $A$ and $B$.
- **PCA Construction:** Utilizes principal components of the pre-trained weights for $A$ and a truncated identity matrix for $B$.

As reported in Table 12, our MROD (SVD-based construction) proves superior, outperforming Random Construction by **1.53%** and PCA Construction by **1.15%** in average accuracy. This performance gap can be attributed to the specific limitations of the alternatives: Random Construction yields vanishingly small gradients that hinder the router's learning, while PCA Construction suffers from significant information loss. In contrast, the SVD-based MROD approach successfully circumvents both issues. It maintains high representational fidelity to pre-trained weights while ensuring sufficient gradient flow for dynamic rank selection.

Table 12: Performance comparison of different rank construction strategies. MROD (SVD-based) achieves the best performance across all datasets by balancing representational fidelity and gradient flow.

| Construction Method | ImageNet-A | ImageNet-R | UCF101 | Avg. |
|---|---|---|---|---|
| Random Construction | 64.03 | 80.96 | 69.36 | 71.45 |
| PCA Construction | 65.04 | 81.50 | 68.99 | 71.83 |
| **MROD (SVD-based)** | **66.15** | **82.47** | **70.31** | **72.98** |

## C.10 ANALYZING ROUTER DYNAMICS AND CONTEXTUAL ADAPTATION IN CONTINUOUS LEARNING

To establish the strength of MuRA's claims under varying data distributions, we conduct an evaluation of sequential distribution shifts (CL-style streams) to validate that CRU appropriately adapts the router over time and visualize how its rank routing evolves as the distribution changes. We define the stream over seven domains, presented sequentially in order of decreasing intrinsic complexity: ImageNet-A, ImageNet-V, ImageNet, ImageNet-R, ImageNet-S, DTD, and EuroSat, with 1,000 steps sampled per domain.

The overall performance comparison (Figure 12) in the continuously shifting TTA scenario shows MuRA with CRU maintaining the highest accuracy, demonstrably superior to MuRA without CRU and the prior state-of-the-art method DPE. This indicates that the CRU strategy provides exceptional robustness not only in single-domain performance but also in continuous multi-domain adaptation.

To further reveal the efficacy of the CRU strategy, we analyze the internal adaptive mechanism. We visualize the internal mechanisms by tracking the router's state changes across the Q, K, V, and O projection matrices. We examine the Router Rank-Utilization Entropy to assess the rank utilization profile of different matrices. The Entropy Analysis (Figure 13) reveals that using CRU leads to a decreasing entropy trend across the overall domain shifts. This confirms the router's successful stabilization and increased certainty in rank selection. The entropy decline of the Q matrix is slow,

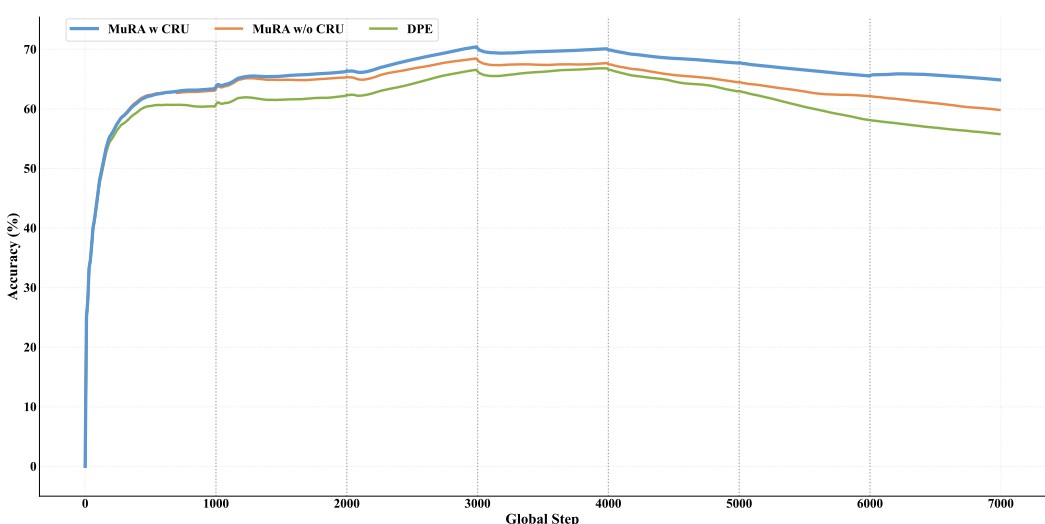

Figure 12: **Overall Performance Baseline in Continuous Learning.** MuRA with CRU consistently outperforms MuRA w/o CRU and DPE baselines.

reflecting its more uniform routing profile and tendency to aggregate knowledge from a broader mix of ranks. In contrast, the K, V, and O projection matrices' entropy declines faster, aligning with their more focused routing profiles.

The Routing Profile Analysis (Figure 14) confirms our central hypothesis by showing that rank selection and allocation adapts to domain complexity. Router activity reveals that in complex domains (e.g., ImageNet-A, ImageNet-V, ImageNet), the router tends to allocate high ranks (Rank 8, 16, 32), confirming the necessity of a larger updating capacity for these significant distributional shifts. Conversely, for less complex domains (ImageNet-R, ImageNet-S, DTD, EuroSat), smaller ranks (Rank 2, 4) are predominantly utilized. The O matrix, which is an output projection matrix and does not participate in the core attention computation, thus consistently favors smaller ranks across all domains, confirming the fine-grained, matrix-specific adaptation of CRU.

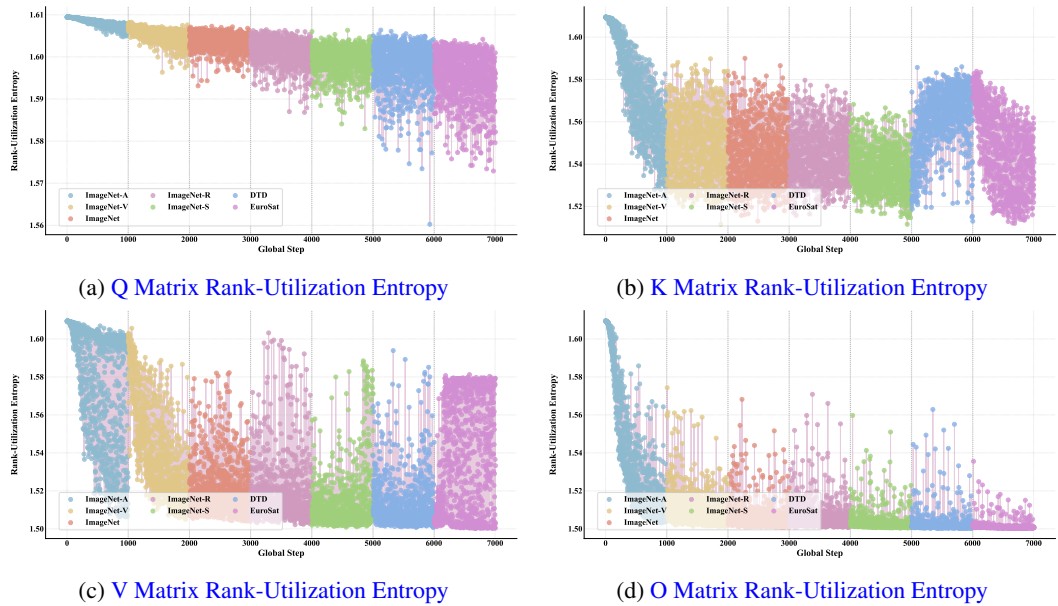

(a) Q Matrix Rank-Utilization Entropy

(b) K Matrix Rank-Utilization Entropy

(c) V Matrix Rank-Utilization Entropy

(d) O Matrix Rank-Utilization Entropy

Figure 13: **Router Rank-Utilization Entropy (Q, K, V, O) across Sequential Domain Shifts.**

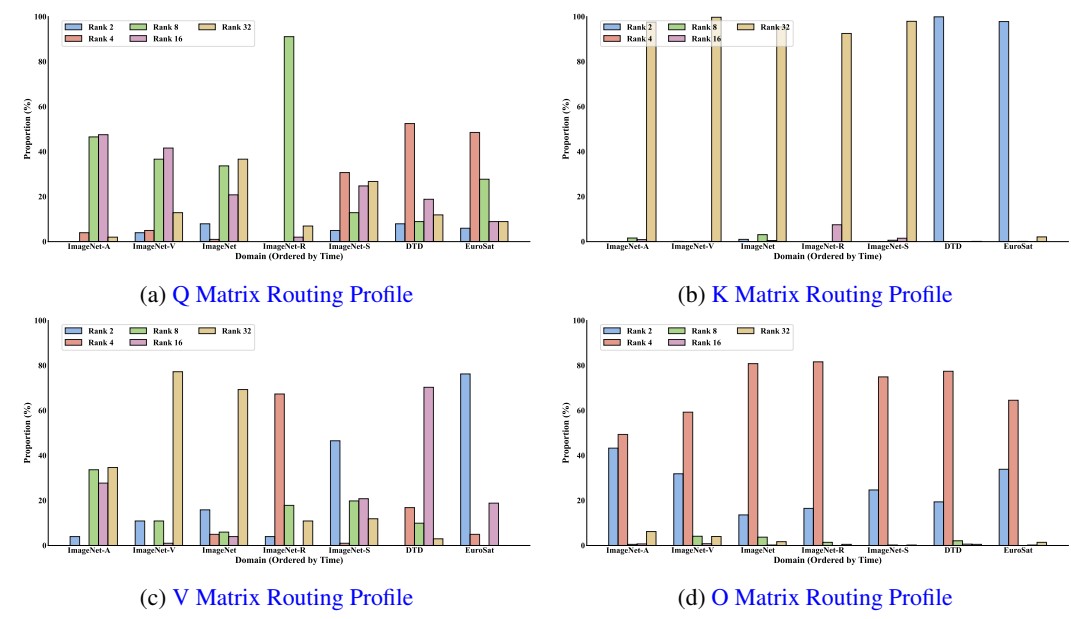

(a) Q Matrix Routing Profile

(b) K Matrix Routing Profile

(c) V Matrix Routing Profile

(d) O Matrix Routing Profile

Figure 14: **Router Routing Profile (Q, K, V, O) across Sequential Domain Shifts.**

## D    THEORETICAL ANALYSIS

We prove that MuRA can exactly recover the original pretrained model's output when the router $W_r$ is initialized with zeros. This property ensures stable performance at the beginning of adaptation by maintaining consistency with the pretrained state. When $W_r$ is initialized to zero, the router's linear projection yields $W_r h_l = \mathbf{0}$. After applying softmax to the zero vector, we obtain a uniform distribution over $k$ components: $\pi_i(h_l) = \frac{1}{k}, \quad \forall i = 1, \ldots, k$. The output can then be derived through the following steps:

$$o = \sum_{i=1}^{k} \pi_i(h_l) A_i B_i^\top h_l + \bar{R} h_l \tag{12}$$

$$= \sum_{i=1}^{k} \frac{1}{k} A_i B_i^\top h_l + \frac{1}{k} \sum_{i=1}^{k} R_i h_l \tag{13}$$

$$= \frac{1}{k} \sum_{i=1}^{k} \left( A_i B_i^\top + R_i \right) h_l \tag{14}$$

$$= \frac{1}{k} \sum_{i=1}^{k} W h_l \quad (\text{since } W = A_i B_i^\top + R_i) \tag{15}$$

$$= W h_l. \tag{16}$$

This derivation shows that MuRA exactly recovers the pretrained transformation $W$, thereby maintaining the original model's behavior at initialization.

## E    THE USE OF LARGE LANGUAGE MODELS (LLMS)

We acknowledge the use of Large Language Models (LLMs) in this work, primarily to assist with:

- Academic writing and prose refinement
- Code documentation and refactoring
- Technical writing revision

These tools were used solely for improving presentation clarity and writing quality, while all technical contributions, experimental designs, and results analysis were conducted independently by the authors.

# F  DATASET DETAILS

## F.1  DATASET DESCRIPTIONS

In Table 13, we present the detailed statistics of each dataset used in our experiments, including the number of classes, sizes of different sets, and their original tasks.

## F.2  TEXTUAL PROMPTS

In Table 14, we detail the specific hand-crafted prompts utilized for each dataset.

Table 13: Detailed statistics of datasets used in experiments. Note that the last 4 ImageNet variant datasets are designed for evaluating model robustness and only contain test sets.

| Dataset | Classes | Training | Validation | Testing | Task |
|---|---|---|---|---|---|
| Caltech101 (Fei-Fei et al., 2004) | 100 | 4,128 | 1,649 | 2,465 | Object recognition |
| DTD (Cimpoi et al., 2014) | 47 | 2,820 | 1,128 | 1,692 | Texture classification |
| EuroSAT (Helber et al., 2019) | 10 | 13,500 | 5,400 | 8,100 | Satellite image classification |
| FGVCAircraft (Maji et al., 2013) | 100 | 3,334 | 3,333 | 3,333 | Fine-grained aircraft recognition |
| Flowers102 (Nilsback & Zisserman, 2008) | 102 | 4,093 | 1,633 | 2,463 | Fine-grained flower recognition |
| Food101 (Bossard et al., 2014) | 101 | 50,500 | 20,200 | 30,300 | Fine-grained food recognition |
| ImageNet (Deng et al., 2009) | 1,000 | 1.28M | - | 50,000 | Image classification |
| OxfordPets (Parkhi et al., 2012) | 37 | 2,944 | 736 | 3,669 | Fine-grained pet recognition |
| StanfordCars (Krause et al., 2013) | 196 | 6,509 | 1,635 | 8,041 | Fine-grained car recognition |
| SUN397 (Xiao et al., 2010) | 397 | 15,880 | 3,970 | 19,850 | Scene recognition |
| UCF101 (Soomro et al., 2012) | 101 | 7,639 | 1,898 | 3,783 | Action recognition |
| ImageNet-V2 (Recht et al., 2019) | 1,000 | - | - | 10,000 | Distribution shift |
| ImageNet-Sketch (Wang et al., 2019) | 1,000 | - | - | 50,889 | Sketch domain adaptation |
| ImageNet-A (Hendrycks et al., 2021b) | 200 | - | - | 7,500 | Natural adversarial examples |
| ImageNet-R (Hendrycks et al., 2021a) | 200 | - | - | 30,000 | Rendition robustness |

Table 14: Textual prompts used in experiments. Following DPE (Zhang et al., 2024a), we use the same textual prompts for fair comparison.

| Dataset | Prompts |
|---|---|
| ImageNet (Deng et al., 2009) | "itap of a {CLASS}." |
| ImageNet-V2 (Recht et al., 2019) | "a bad photo of the {CLASS}." |
| ImageNet-Sketch (Wang et al., 2019) | "a sketch of {CLASS}." |
| ImageNet-A (Hendrycks et al., 2021b) | "a photo of the large {CLASS}." |
| ImageNet-R (Hendrycks et al., 2021a) | "a {CLASS} in a video game." |
| | "art of the {CLASS}." |
| | "a photo of the small {CLASS}." |
| Caltech101 (Fei-Fei et al., 2004) | "a photo of a {CLASS}." |
| DTD (Cimpoi et al., 2014) | "{CLASS} texture." |
| EuroSAT (Helber et al., 2019) | "a satellite photo of {CLASS}." |
| FGVCAircraft (Maji et al., 2013) | "a photo of a {CLASS} aircraft." |
| Flowers102 (Nilsback & Zisserman, 2008) | "a photo of a {CLASS} flower." |
| Food101 (Bossard et al., 2014) | "a photo of {CLASS} food." |
| OxfordPets (Parkhi et al., 2012) | "a photo of a {CLASS} pet." |
| StanfordCars (Krause et al., 2013) | "a photo of a {CLASS}." |
| SUN397 (Xiao et al., 2010) | "a photo of a {CLASS}." |
| UCF101 (Soomro et al., 2012) | "a person doing {CLASS}." |

