# OpenReview forum: "MuRA: Multi-Rank Adaptation for Efficient and Effective Test-Time Vision-Language Generalization"
_ICLR.cc/2026/Conference — Submitted to ICLR 2026_

### Official Review · Reviewer_nRHo · 2025-10-21

**Soundness:** 2
**Presentation:** 1
**Contribution:** 3
**Rating:** 4
**Confidence:** 5

**Summary:**

The paper targets knowledge-adaptive TTA for VLMs and argues that a static LoRA rank is suboptimal because different inputs/datasets require different adaptation capacity. Empirically, the optimal rank varies widely across domains and correlates strongly with image entropy. The proposed MuRA prepares multiple rank-specific LoRA modules via Multi-Rank Orthogonal Decomposition (MROD) and soft, token-level routing (UCF) with Continuous Router Updating (CRU). MuRA delivers state-of-the-art average accuracy on ImageNet OOD and cross-domain suites with attractive accuracy–efficiency trade-offs.

**Strengths:**

- Clear diagnosis & evidence. The paper convincingly shows rank sensitivity across datasets and its linear relation to image entropy, motivating dynamic rank selection.

- Well-designed, cohesive method. MROD: principled SVD-based init yielding orthogonal residuals; improves stability of one-step TTA. UCF (token-level soft MoE) + CRU: learns token-wise rank preferences over time; soft routing > hard, token-level > instance-level (especially with CRU).

**Weaknesses:**

- Varying data distributions. Although the paper claims strength under varying data distributions, current experiments use a single test distribution per benchmark. Please evaluate sequential distribution shifts (CL-style streams) to show that CRU adapts appropriately over time and visualize how CRU’s rank routing evolves as the distribution changes. Framing the method explicitly as strong for CL + TTA would also sharpen the contribution.

- Academic formatting quality. The manuscript’s presentation needs polishing (e.g., oversized figures, occasional image blurring/pixelation, inconsistent layout). Please standardize figure sizes/resolution, ensure vector graphics where possible for professional readability.

**Questions:**

- Please visualize the evolution of CRU’s rank routing as the distribution changes (e.g., rank-utilization entropy over the stream, per-domain routing profiles) --- See Weakness 1.
- Please improve the paper formatting quality --- See Weakness 2.

---

> ### Author Response · Authors · 2025-11-24
> **Author Response (1/1)**
>
> We thank Reviewer nRHo for the thoughtful review and valuable suggestions. We appreciate the recognition of our work as a "well-designed, cohesive method" with "clear diagnosis & evidence." We are encouraged by the positive assessment of MROD, UCF, and CRU. We address the specific concerns below.
>
> **Weaknesses**
>
> **Robustness under Continuous Distribution Shifts**
>
> We thank the reviewer for this insightful suggestion. To strengthen the claim of our method's robustness, we conducted a Continuous Learning (CL) style evaluation and visualized the router's evolution.
>
> We defined a sequential stream over seven domains, ordered by decreasing intrinsic complexity: ImageNet-A $\to$ ImageNet-V $\to$ ImageNet $\to$ ImageNet-R $\to$ ImageNet-S $\to$ DTD $\to$ EuroSat. The model adapts continuously for 1,000 steps per domain.
>
> As shown in Figure 12 of the revised paper, MuRA with CRU consistently maintains the highest accuracy throughout the stream. This indicates that the CRU strategy provides exceptional robustness not only in single-domain performance but also in continuous multi-domain adaptation.
>
> To show how CRU’s rank routing evolves as the distribution changes, we provide two complementary analyses in the revised manuscript:
> * **Router Rank-Utilization Entropy analysis (Figure 13 in revised paper):** We track the rank-utilization entropy to measure routing uncertainty. The consistent downward trend confirms gradual stabilization from exploration to confident exploitation, refuting the concern of aimless drift. Notably, the distinct decay patterns observed across the Q, K, V, and O matrices directly result in differentiated routing profiles for each projection layer.
> * **Routing Profile analysis (Figure 14 in revised paper):** We visualize the actual rank proportions selected per domain. The results confirm that the routing is semantically adaptive: the router intelligently allocates high ranks (e.g., 16, 32) for complex domains like ImageNet-A, while shifting preference to lower ranks (e.g., 2, 4) for simpler domains like EuroSat. Specifically, the O matrix consistently favors smaller ranks, reflecting its lower capacity requirement as an output projection excluded from core attention.
>
> **Presentation and Formatting Improvements**
>
> Regarding the figure quality, we clarify that the figures in the initial submission were indeed vector graphics; however, the text elements had been converted into non-selectable outlines. We have resolved this issue in the revised manuscript, thereby improving text accessibility without altering the visual content.
>
> To enhance readability, we have implemented the following layout improvements:
> * We have split the original Figure 1 into two distinct figures. Revised Figure 1 now focuses on the efficiency-accuracy trade-off (augmented with a new Radar Chart to highlight MuRA's superiority), while Revised Figure 2 illustrates the methodological comparison.
> * The original Figure 2 (now Revised Figure 3) has been integrated using a wrap-figure layout to optimize space utilization.
> * We have also refined the alignment of the subfigures in the original Figure 4 (now Revised Figure 5) to ensure consistent heights.
>
> **Questions**
>
> **Questions regarding CRU evolution and Formatting**
>
> Please refer to our responses in **Robustness under Continuous Distribution Shifts** and **Presentation and Formatting Improvements** above, where we provide the requested visualizations and formatting details.

---

### Official Review · Reviewer_iFPH · 2025-10-29

**Soundness:** 2
**Presentation:** 3
**Contribution:** 2
**Rating:** 4
**Confidence:** 4

**Summary:**

The paper proposes MuRA, a test-time adaptation method for VLMs that fuses multiple LoRA modules of different ranks via a softmax-weighted router and initializes the rank components through an SVD-based “Multi-Rank Orthogonal Decomposition” (MROD). A “Continuous Router Updating” (CRU) strategy is claimed to retain routing knowledge across samples. The method adapts only the deepest visual layer and uses 63 augmentations per test image with entropy-weighted selection. Experiments report gains on ImageNet variants and cross-domain datasets.

**Strengths:**

1. Practical, lightweight design: restricting adaptation to the deepest layer is reasonable for efficiency and simplicity.
2. Competitive results on multiple benchmarks, with ablations that partially justify the components (MROD, UCF, CRU).
3. Clear framing of rank sensitivity and correlation with image entropy; the motivation for dynamic rank choice is intuitive.

**Weaknesses:**

1. Limited technical novelty: the core idea—combining multiple LoRA ranks with a softmax router—is a straightforward mixture-of-experts/gating over adapters and feels incremental relative to existing PEFT/TTA adapter ensembles.
2. SVD vs. random partition: the paper does not convincingly demonstrate why SVD-based rank construction is superior to simpler alternatives (e.g., random splits, fixed-rank LoRA, or PCA variants). Please add controlled comparisons (same parameter budget) and report effect sizes.
3. CRU under-specified: the Continuous Router Updating component (around Line 300) is described in one sentence without algorithmic detail. How exactly is the router state retained/reset across samples/batches? What regularization, optimizer, learning rate schedule, and stability safeguards are used? Provide pseudo-code and a failure/sensitivity analysis.
4. Efficiency concerns: generating 63 augmentations per test image can be costly. The paper should include a clear time and memory complexity analysis (asymptotics and wall-clock),  and throughput vs. accuracy trade-offs (e.g., 8/16/32/63 views). Report results with fewer views to show robustness.

**Questions:**

See weaknesses.

---

> ### Author Response · Authors · 2025-11-24
> **Author Response (1/2)**
>
> We thank Reviewer iFPH for the constructive feedback. We appreciate the recognition of our "practical, lightweight design" and "competitive results". We address the specific concerns below.
>
> **Weaknesses**
>
> **Clarification on Technical Novelty**
>
> We are the first to identify that dynamic rank adaptation is pivotal for Test-Time Adaptation (TTA). We demonstrate that our multi-rank Mixture-of-Experts (MoE) framework provides a systematic and effective solution to this challenge. Furthermore, our specialized mechanisms—specifically the Rank-adaptive MoE framework, Inference-efficient architecture, Sample-shared knowledge learning, and Stabilized Initialization—underpin MuRA’s comprehensive superiority in efficiency, robustness, and accuracy.
>
> In summary, the Mixture-of-Experts system constructed in MuRA is characterized by the following distinct features:
>
> * **Rank-adaptive MoE framework:** To adaptively allocate adapters with adequate capacity for each input complexity, we introduce an MoE framework featuring diverse rank combinations. Figure 6a demonstrates the superiority of our multi-rank system over homogeneous (same-rank) MoE designs, while Table 3 validates that MuRA significantly outperforms single-rank LoRA TTA methods with comparable parameter budgets.
> * **Inference-efficient architecture:** To meet the stringent inference efficiency requirements of TTA, we optimize the number and insertion position of the MoE layers. As shown in Figure 6b & 6c, Figure 1, and Figure 11, this targeted design of adapting only the deepest layer achieves a superior balance between effectiveness and efficiency.
> * **Sample-shared knowledge learning:** To exploit sample-shared knowledge in the TTA scenario, we introduced the Continuous Router Updating (CRU) mechanism to continuously accumulate parameter updates across input samples, acquiring a robust "Complexity-Capacity Mapping". The effectiveness of CRU can be justified by Table 3, Figure 8, Figure 10, and Figure 12.
> * **Stabilized Initialization:** To enrich the pre-trained knowledge for different experts and stabilize the TTA optimization process, we adopted the Multi-Rank Orthogonal Decomposition (MROD) initialization and show its efficacy in Figure 5 and Table 3.
>
> To further clarify these technical innovations, we have comprehensively supplemented the motivation for the CRU strategy in Section 3.2 and summarized the points above in the Conclusion section of the revised paper.
>
> **Effectiveness of SVD-based Rank Construction**
>
> Actually, the SVD-based approach is the most effective design for our architecture. As demonstrated in Figures 5a and 5b, this approach provides superior initial knowledge representation and optimization stability.
>
> To further empirically validate this choice, we conducted a controlled comparison against two alternatives, ensuring all methods share an identical parameter budget and initialize the backbone by aggregating rank-specific residuals.
> * **Random Construction:** Using Kaiming uniform initialization for $A$ and $B$.
> * **PCA Construction:** Using principal components of the pre-trained weights for $A$ and a truncated identity matrix for $B$.
>
> The SVD-based construction proves superior by average accuracy (+1.53% vs. Random, +1.15% vs. PCA). This can be attributed to the fact that Random Construction yields vanishingly small gradients that hinder the router's learning, and PCA Construction suffers from significant information loss. However, the SVD-based approach successfully circumvents both issues. It maintains high representational fidelity to pre-trained weights while ensuring sufficient gradient flow for dynamic rank selection.
>
> | Method | ImageNet-A | ImageNet-R | UCF101 | Avg. |
> | :--- | :---: | :---: | :---: | :---: |
> | **SVD-based** | **66.15** | **82.47** | **70.31** | **72.98** |
> | Random Construction | 64.03 | 80.96 | 69.36 | 71.45 |
> | PCA Construction | 65.04 | 81.50 | 68.99 | 71.84 |

---

> ### Author Response · Authors · 2025-11-24
> **Author Response (2/2)**
>
> **CRU Mechanism and Implementation Details**
>
> * **Router State Retention & Implementation Details:** The router state ($W\_r$) persists across the entire TTA stream and is updated continuously without resetting to the initial state. Given the minimal parameterization of the router projection matrix $W_r \in \mathbb{R}^{d \times K}$ (specifically $768 \times 5$), we find that a standard constant learning rate is sufficient for stable convergence without requiring complex schedulers or additional regularization.
> * **Pseudo-code:** To clearly illustrate the CRU execution process, we have explicitly detailed the steps—including the specific retention and update logic—in Algorithm 1. Specifically, lines 8-10 describe the continuous router updating while adapters are reset.
> * **Sensitivity Analysis:** Actually, the CRU strategy is critical to our framework. To validate this design, we analyzed the impact of forcing router resets at different frequencies, as shown in Figure 10. The results demonstrate that continuous updating (Reset Times = 0) yields the highest accuracy. Conversely, any interruption to this continuity leads to performance degradation. As the reset frequency increases, accuracy drops significantly and eventually converges to the "w/o CRU" baseline. This confirms that accumulating unbroken router context is essential for performance.
>
> **Efficiency and Robustness Analysis**
>
> We initially adopted the 64-view setting (63 augmented + 1 original) to ensure a fair comparison and consistency with prior state-of-the-art protocols (e.g., TPT, DPE, TTL).
>
> To demonstrate MuRA's robustness and efficiency across varying view settings, we conducted a comprehensive analysis of Throughput, Memory Usage, and Accuracy under different view counts ($8, 16, 32, 64$). These results are presented in Figure 11 and summarized below:
>
> * **Observations & Analysis:** As the results indicate, MuRA consistently maintains SOTA accuracy while achieving the highest throughput and the lowest memory footprint across all view settings.
> * **Robustness:** This stability stems from MuRA's ability to dynamically select the optimal adaptation capacity, thereby minimizing reliance on extensive multi-view statistics. In contrast, methods like DPE rely heavily on aggregating statistics across numerous views; consequently, their performance collapses significantly when views are scarce.
> * **Efficiency:** MuRA’s efficiency originates from our architectural design—specifically, restricting multi-rank parameters to the deepest layer—which significantly minimizes computational and memory overhead.

---

### Official Review · Reviewer_2MGn · 2025-10-30

**Soundness:** 3
**Presentation:** 3
**Contribution:** 3
**Rating:** 6
**Confidence:** 4

**Summary:**

This paper proposes MuRA, a test-time adaptation (TTA) method for CLIP that addresses the limitations of static rank configurations in prior knowledge adaptation paradigms. The core idea is that visual inputs with varying information density may require different adaptation capacities. To this end, the method dynamically selects and fuses multiple low-rank adaptation components to achieve efficient adaptation across diverse image types. Extensive experiments on 15 datasets demonstrate competitive performance.

**Strengths:**

- The paper is well-written and easy to understand, with clear explanations of the method and experimental results.
- The knowledge adaptation paradigm represents an interesting form of TTA.

**Weaknesses:**

- Intrinsic limitation of the knowledge adaptation paradigm.   The proposed method appears tightly coupled with a specific architecture and may not generalize well to other baselines such as CLIP with a ResNet-50 backbone.
- Risk of overconfidence from entropy minimization. The use of entropy minimization loss may cause over-confident predictions during test-time adaptation, which could negatively affect model calibration.
- The manuscript lacks comparisons with the state-of-the-art VLM TTA methods, such as MCP[1], GS-Bias[2], and TT-RAA[3].

[1] Multi-Cache enhanced Prototype Learning for Test-Time Generalization of Vision-Language Models. ICCV 2025

[2] GS-Bias: Global-Spatial Bias Learner for Single-Image Test-Time Adaptation of Vision-Language Models. ICML 2025

[3] Test-Time Retrieval-Augmented Adaptation for Vision-Language Models. ICCV 2025

**Questions:**

- In Table 3, the performance improvement of Unified Component Fusion (UCF) after MROD appears marginal.   Does this suggest that the contribution of UCF is limited?
- What theoretical justification supports the design of the Continuous Router Updating (CRU) strategy? I am concerned that continuously updating the router across test samples may lead to the accumulation of adaptation errors or drift over time.
- The paper lacks visualization or interpretability studies (e.g., t-SNE feature visualizations) that could provide insights into how the proposed adaptation influences the representation space.

---

> ### Author Response · Authors · 2025-11-24
> **Author Response (1/2)**
>
> We sincerely thank Reviewer 2MGn for the encouraging feedback and for recognizing our paper as "well-written" and our method as an "interesting form of TTA." We appreciate the positive evaluation of our extensive experiments and the recognition of our contribution. We address the specific concerns below.
>
> **Weaknesses**
>
> **Applicability to Other Architectures**
>
> While we implemented MuRA on CLIP (ViT) due to its popularity and for fair comparison with prior work, the MuRA architecture is inherently architecture-agnostic. MuRA essentially models the update of different ranks for a dense layer, making it applicable to any dense layer (Linear or Convolutional). We believe extending MuRA to models like ResNet or large language models (LLMs) would be straightforward. Due to time constraints, we were unable to extend MuRA to these additional frameworks. We propose exploring MuRA's properties and applicability across a wider range of architectures as a promising direction for future work.
>
> **Mitigation of Overconfidence**
>
> We acknowledge that the entropy minimization loss utilized by prior work, such as DPE and TPT, poses an inherent risk of overconfidence. However, MuRA incorporates an adaptive weighting scheme, originally proposed by TTL, which is verified effective in Appendix Table 7 (Eq. 2). This mechanism leverages prediction uncertainty to mitigate the risk: high-entropy (uncertain) samples are down-weighted in the loss calculation, thereby preventing the model from forcefully adapting to ambiguous samples.
>
> **Comparisons with Recent State-of-the-Art VLM TTA Methods**
>
> We sincerely thank the reviewer for pointing out these recent state-of-the-art test-time adaptation methods for vision-language models. After incorporating comparisons with these recent methods, MuRA remains the new state-of-the-art.
>
> As shown in the table below, MuRA achieves the highest average accuracy and OOD average accuracy. On particularly difficult datasets such as ImageNet-A and ImageNet-R, MuRA demonstrates substantial gains—+5.56% over TT-RAA on ImageNet-A and +1.89% over MCP++ on ImageNet-R. These results highlight MuRA’s strong capability in handling extreme domain shifts, attributable to its dynamic multi-rank adaptation mechanism.
>
> We commit to including these updated comparisons in both the main experiments section and related work of the revised manuscript, along with proper citations, to ensure a fair, complete, and up-to-date evaluation of MuRA’s performance.
>
> | Method | ImageNet | ImageNet-A | ImageNet-V2 | ImageNet-R | ImageNet-S | OOD Avg. | Average |
> | :--- | :---: | :---: | :---: | :---: | :---: | :---: | :---: |
> | MCP | 72.37 | 59.71 | 65.16 | 81.59 | 53.89 | 65.09 | 66.54 |
> | MCP++ | **72.64** | 59.78 | 65.77 | 81.73 | **54.39** | 65.42 | 66.86 |
> | GS-Bias | 69.02 | 54.55 | 63.37 | 76.64 | 48.21 | 60.69 | 62.36 |
> | GS-Bias + E. | 70.57 | 56.61 | 64.62 | 80.49 | 50.33 | 63.01 | 64.52 |
> | TT-RAA | — | 60.59 | 64.69 | 80.58 | 49.98 | 63.96 | — |
> | **MuRA (Ours)** | 72.46 | **66.15** | **66.48** | **82.47** | 51.76 | **66.72** | **67.86** |

---

> ### Author Response · Authors · 2025-11-24
> **Author Response (2/2)**
>
> **Questions**
>
> **Contribution of Unified Component Fusion (UCF)**
>
> We clarify that UCF's contribution is exceptionally promising and foundational. UCF serves as the structural prerequisite that enables the Continuous Router Updating (CRU) strategy. Without the fusion mechanism provided by UCF, dynamic rank routing (CRU) would be impossible to implement. As evidenced in Table 3 of our manuscript, applying CRU on top of UCF triggers a substantial performance jump to 72.98%, with significant gains on challenging shifts like ImageNet-A (+2.23%).
>
> **Theoretical Justification for Continuous Router Updating (CRU)**
>
> **1. Overview of CRU Strategy**
> To briefly recapitulate, Continuous Router Updating (CRU) is a mechanism where the router's parameters are maintained and updated continuously across different test samples, whereas the knowledge adaptive modules (LoRA) are reset to their initialized state after each sample.
>
> **2. Core Rationale: Structural vs. Semantic Adaptation**
> The theoretical foundation of CRU lies in the distinction between Semantic Adaptation and Structural Adaptation.
> * **LoRA targets Semantic Adaptation:** It adjusts feature representations to align with specific class semantics. Continuous updates in this manner risk overfitting to incorrect pseudo-labels, leading to rapid error accumulation.
> * **The Router targets Structural Adaptation:** Conversely, the router learns a "Complexity-Capacity Mapping." This represents universal meta-knowledge that determines the optimal representational capacity (rank) required by the input's complexity. Crucially, this structural mapping is orthogonal to specific class labels, making the router's continuous evolution robust to individual misclassifications.
>
> **3. Detailed Justification**
> * **Why LoRA Parameters Must Be Reset (The Risk of Semantic Error Accumulation):** LoRA parameters ($\theta_{\text{LoRA}}$) directly manipulate semantic features to minimize prediction entropy ($L_{\text{ent}}$). In unsupervised TTA, if the model initially misclassifies a sample (predicting $y_{\text{wrong}}$), the entropy minimization objective will push the weights to increase confidence in this incorrect prediction ($\theta_{\text{LoRA}} \leftarrow \theta_{\text{LoRA}} - \eta \nabla_{\theta} L_{\text{ent}}$). This injects erroneous semantic bias. If not reset, this bias propagates to subsequent samples, causing catastrophic forgetting of pre-trained knowledge.
> * **Why CRU is Safe and Beneficial (The Stability of Structural Adaptation):** Unlike LoRA, the Router ($W_r$) does not learn which class is correct, but rather determines the appropriate capacity required to accommodate the complexity of the current input.
>     * **Decoupling Correctness from Capacity:** As shown in our analysis (Figure 3), the optimal rank is highly correlated ($R^2=0.913$) with the input's visual complexity (image entropy).
>     * **Robustness to Misclassification:** Even in cases of misclassification, the gradient signal remains structurally valid, indicating that "a specific rank was needed to sharpen the distribution for this input complexity." This "Complexity-to-Rank" mapping constitutes stable meta-knowledge that does not suffer from the drift inherent in semantic features.
>
> * **Empirical Validation:**
>   * **Sensitivity Analysis:** Figure 10 demonstrates that resetting the router ($Reset Times > 0$) consistently degrades performance compared to continuous updating ($Reset Times = 0$), confirming that the accumulated routing history yields a net benefit.
>   * **Continuous Learning Stability:** In sequential domain shift experiments (Figure 12), MuRA equipped with CRU maintains high accuracy over time, avoiding the performance collapse observed in methods relying on semantic memory (e.g., DPE).
>
> **Visualization Study**
>
> Detailed visualizations had been already provided in Appendix Figure 6 and Figure 7 of our original submission (now in Figure 7 and Figure 8 of the revised manuscript). To improve visibility, we have moved them to the main text in the revised manuscript.
>
> * **Figure 6 (Attention Analysis)** demonstrates that low-rank components tend to generate diffuse attention patterns suitable for processing low-information-density inputs, while high-rank components produce concentrated attention essential for precisely capturing targets in complex inputs. Crucially, it shows that MuRA is capable of dynamically selecting and integrating these components, thereby synthesizing a superior attention map tailored to the specific input.
> * **Figure 7 (Rank Preference Analysis)** displays token-level rank preference maps generated by the router. It demonstrates that—especially with the Continuous Router Updating (CRU) strategy—rank selection becomes highly semantic, exhibiting consistent choices across semantically similar regions (e.g., background vs. foreground), thereby effectively refining the input-to-rank mapping.

---

> ### Comment · Reviewer_2MGn · 2025-11-27
> **Official Comment by Reviewer 2MGn**
>
> I thank the authors for their responses, which address my questions. I'll keep my current rating.

---

> > ### Author Response · Authors · 2025-11-27
> > **Thanks for reviewer 2MGn's response**
> >
> > We sincerely thank Reviewer 2MGn for confirming that our responses have successfully addressed the questions. We appreciate the time and effort dedicated to reviewing our work.

---

### Official Review · Reviewer_AXZk · 2025-11-01

**Soundness:** 2
**Presentation:** 2
**Contribution:** 2
**Rating:** 4
**Confidence:** 5

**Summary:**

This paper proposes MuRA (Multi-Rank Adaptation), a test-time adaptation (TTA) framework designed for Vision-Language Models (VLMs) such as CLIP. MuRA adapts both visual and textual embeddings across multiple ranks in a unified optimization objective, yielding more flexible and robust test-time updates. Experiments on benchmark datasets show improved accuracy and robustness compared to single-rank or fixed adapter baselines.

**Strengths:**

S1. **Comprehensive Experiments.**
The paper adheres to the standard evaluation protocols established in the VLM TTA community and shows consistent performance gains over prior baselines across multiple benchmark datasets.

S2. **Balanced technical depth and clarity.**
The paper provides clear algorithmic exposition, ablation studies on rank configurations, and qualitative analyses supporting the motivation in Appendix.

**Weaknesses:**

W1. **Missing comparison against multiple same-rank adapters.**
While the idea of employing multiple ranks for adaptation is interesting, the paper does not convincingly show that the improvement comes from rank diversity itself rather than from simply using multiple adapters. The ablation study compares only single-rank vs. multi-rank configurations, but there is **no baseline that uses several adapters of the same rank. Without this comparison, it remains unclear whether MuRA’s advantage originates from heterogeneous rank composition or just increased model capacity. Including such an experiment would make the contribution much more convincing.

W2. **Unfair comparison due to unmatched adaptation capacity.**
Several reported gains may stem from larger trainable capacity rather than the proposed multi-rank design. In many tables (Table 1&2), MuRA appears to use more total trainable parameters than baselines that use a single adapter of a fixed rank or adapt only one branch. Without capacity-controlled baselines, the comparison is confounded. To ensure the fairness, the paper should report the total trainable parameters, further match total trainable parameters.

W3. **Inference Overhead.**
Unlike prior single-LoRA approaches, MuRA introduces a gating module that determines which LoRA branch to activate during inference.
In practice, this requires computing the forward pass of both the base model and a LoRA module selected by gating module, followed by their weighted summation. This design inevitably incurs additional inference overhead compared to a single LoRA model, where the LoRA weights can be merged into the base weights, resulting in identical forward cost. Therefore, the authors should report and analyze the inference-time computational cost of MuRA, including latency, FLOPs, and throughput, to clarify the trade-off between performance gain and efficiency.

W4. **Incremental performance gains.**
As shown in Tables 2–4, the proposed method achieves approximately 1.0–2.5% improvement over single-rank baselines. While the gains are consistent, they are relatively modest given the additional complexity introduced by the multi-rank design. Considering that MuRA requires multiple adapters and a gating mechanism at inference, the improvement-to-overhead ratio appears limited. A more detailed efficiency analysis or scenarios where MuRA provides significantly larger benefits (e.g., under extreme domain shifts) would help justify the practical value of the approach.

**Questions:**

I wrote all my concerns in Weakness section.

---

> ### Author Response · Authors · 2025-11-24
> **Author Response (1/2)**
>
> We thank Reviewer AXZk for the constructive feedback and insightful comments. We appreciate the recognition of our comprehensive experiments and the clarity of our technical depth. We hope to address the concerns of the reviewer with the responses below.
>
> **Weaknesses**
>
> **Comparisons against multiple same-rank adapters**
>
> The comparison had already been presented in our original submission as Figure 5(a) (now updated to Figure 6(a) in the revised manuscript), where the dashed lines corresponded exactly to the baselines using "multiple same-rank adapters" (specifically configurations of \{2, 2, 2, 2, 2\} and \{32, 32, 32, 32, 32\}). To enhance readability, we have now explicitly labeled these baselines in the revised paper.
>
> As illustrated in Figure 6(a), the "multiple same-rank adapters" configurations (both low-rank \{2,...\} and high-rank \{32,...\}) consistently underperform MuRA’s heterogeneous configuration \{2,4,8,16,32\}, despite having comparable or even greater total parameter counts. This empirical evidence confirms that the performance gain primarily stems from the rank diversity and dynamic selection rather than merely using multiple adapters.
>
> **Matched adaptation capacity comparison**
>
> Our advantages persist even under a parameter-matched comparison. In the table below, we present the parameter counts for the default settings of MuRA and TTL, and introduce a new capacity-matched baseline, $\text{TTL}_{\text{large}}$, which employs uniformly enlarged static ranks to closely align with MuRA’s parameter budget.
>
> 1.  As shown in the table, MuRA consistently outperforms $\text{TTL}_{\text{large}}$. This demonstrates that MuRA’s performance gains stem from the proposed multi-rank design rather than simply larger trainable capacity.
> 2.  Furthermore, it is notable that MuRA outperforms both the default TTL and $\text{TTL}_{\text{large}}$ even with a significantly reduced parameter budget of 0.09M (using rank configuration \{2, 4, 8\}), which further corroborates the critical importance of rank diversity in test-time adaptation scenarios.
>
> | Method | Trainable Params | ImageNet-A | ImageNet-R | UCF101 | Avg. |
> | :--- | :---: | :---: | :---: | :---: | :---: |
> | **MuRA$_{\{2, 4, 8, 16, 32\}}$ (Ours)** | 0.39M | **66.15** | **82.47** | **70.31** | **72.98** |
> | *MuRA$_{\{2, 4, 8\}}$ (Small)* | *0.09M* | *63.28* | *82.09* | *69.71* | *71.69* |
> | $\text{TTL}_{\text{large}}$ (Matched) | 0.38M | 58.51 | 77.60 | 68.52 | 68.21 |
> | TTL (Default) | 0.15M | 60.51 | 77.54 | 69.20 | 69.08 |
>
> **Inference Overhead Analysis**
>
> We had previously illustrated the memory usage, throughput, and performance comparisons between MuRA and the baseline TTL (Single LoRA) in Figure 1(e) (now updated to Figure 1 (Right) in the revised manuscript). To provide a more comprehensive assessment of inference overhead, we have now incorporated FLOPs metrics and expanded the comparison in the table below.
>
> The results demonstrate that MuRA achieves the highest efficiency in terms of memory footprint and throughput while maintaining superior OOD performance. Regarding FLOPs, the difference between MuRA and TTL is marginal, and when compared to other TTA methods such as TPT, this difference is negligible. MuRA's excellent efficiency-performance trade-off stems from its strategic restriction of adaptation to the deepest layer. This design minimizes the gradient backpropagation path, thereby simultaneously optimizing computational throughput and memory footprint without compromising robust performance.
>
> | Method | FLOPs | Memory | Throughput | OOD Avg |
> | :--- | :--- | :--- | :--- | :--- |
> | **MuRA** | 2.4 TFLOPS | **2.05 GB** | **11.26 samples/s** | **66.72** |
> | TTL | 2.25 TFLOPS | 2.45 GB | 6.04 samples/s | 62.80 |
> | TPT | 3.32 TFLOPS | 4.34 GB | 3.2 samples/s | 60.81 |

---

> ### Author Response · Authors · 2025-11-24
> **Author Response (2/2)**
>
> **Incremental performance gains and practical value**
>
> Our MuRA demonstrates promising performance, exhibiting not only superior generalization capabilities under extreme domain shift scenarios but also enhanced scalability as the backbone model size increases. These results further validate the efficacy of our approach. We detail these advantages as follows:
>
> 1.  **Superior Generalization under Extreme Domain Shifts**: Actually, in Tab. 1 & 2 of the main paper, our MuRA has exhibited superior generalization ability in scenarios with extreme domain shift, such as ImageNet-A, DTD, Aircraft, and EuroSat. Concretely, in these challenging cases, the accuracy of native CLIP collapses to below 50%. While, our MuRA outperforms the baseline TTL by substantial margins ranging from 5.6% to 15%.
> 2.  **Superior Scalability with Larger Backbones**: Furthermore, in Appendix C.1 (Table 6) of our revised paper (shown in the table below), our MuRA also demonstrates superior scalability. As shown in the table, MuRA outperforms TTL by +16.75% on the challenging ImageNet-A, and leads by +5.81% and +6.51% on the Overall Average and OOD Average, respectively.
>
> | Method | ImageNet | ImageNet-A | ImageNet-V2 | ImageNet-R | ImageNet-S | Avg | OOD Avg. |
> | :--- | :---: | :---: | :---: | :---: | :---: | :---: | :---: |
> | CLIP-ViT-L/14 | 74.04 | 53.88 | 67.69 | 87.42 | 63.18 | 69.31 | 68.13 |
> | TDA | 76.28 | 61.27 | 68.42 | 88.41 | 64.67 | 71.81 | 70.69 |
> | DPE | 77.87 | 61.09 | 70.83 | 89.18 | **66.33** | 73.06 | 71.86 |
> | TTL | 75.62 | 65.21 | 69.43 | 87.46 | 63.51 | 72.25 | 71.40 |
> | **MuRA (Ours)** | **78.66** | **81.96** | **73.19** | **91.79** | 64.69 | **78.06** | **77.91** |

---

> > ### Comment · Reviewer_AXZk · 2025-11-26
> > **Response by Reviewer AXZk**
> >
> > Thank you for your detailed response to my initial concerns. While most of my concerns have been addressed, I still have a few questions regarding the experimental results.
> >
> > ### Performance discrepancy between TTL_large (Matched) and TTL (Default)
> > It is my understanding that TTL_large increased the rank to match the number of trainable parameters in MuRA. However, the results indicate that TTL (Default), which has a lower rank, actually outperforms the larger-rank version. Could you explain the potential reason for this observation? Additionally, I am curious to know what the results would look like if the parameter count of TTL were explicitly matched to 0.09M.
> >
> > ### Questions regarding the inference overhead analysis
> > Intuitively, one would expect TTL to be more cost-efficient than MuRA since it operates as a single LoRA. However, the provided table suggests that MuRA is superior in terms of both efficiency and performance. Is this outcome primarily driven by the specific rank settings? I am interested to see how the performance would change if the rank of TTL were set to match either the average or the minimum rank of MuRA.

---

> ### Author Response · Authors · 2025-11-27
> **Response to Reviewer AXZk's Thoughtful Questions (Part 1/2)**
>
> We thank Reviewer AXZk for the continued engagement and for raising these thoughtful follow-up questions. We are glad to hear that our previous response addressed most of the initial concerns. Below, we provide the additional experimental results and analyses to address the Reviewer's questions.
>
> **1. Performance discrepancy between TTL_large (Matched) and TTL (Default)**
>
> The reviewer is correct that for $\text{TTL}_{\text{large}}$, we explicitly increased the rank to 42 to align with MuRA's parameter budget. As illustrated in Figure 3 of our revised paper, inputs with varying complexity naturally require different adaptation capacities (ranks). Concretely, a rank that is too low lacks the capacity for sufficient updates, while a rank that is too high leads to excessive updates and potential overfitting on the test sample. Consequently, single-rank methods like TTL degrade when the chosen rank is either excessively large or insufficiently small.
>
> To further validate this, we performed the experiment requested by the reviewer with $\text{TTL}_{\text{small}}$ (Rank=10, 0.09M params) and compared it against both TTL variants and MuRA variants. The results demonstrate that an insufficiently small rank (Rank=10) similarly leads to a performance decline. In contrast, MuRA is not constrained by such single rank limitations. By adaptively providing rank combinations tailored to specific inputs, MuRA effectively unlocks the model's full adaptation potential during the TTA process. This enables MuRA to achieve superior performance even with a significantly reduced parameter budget (0.09M).
>
> | Method | Trainable Params | ImageNet-A | ImageNet-R | UCF101 | Avg. |
> | :--- | :--- | :--- | :--- | :--- | :--- |
> | **MuRA$_{\{2, 4, 8, 16, 32\}}$ (Ours)** | 0.39M | **66.15** | **82.47** | **70.31** | **72.98** |
> | *MuRA $_{\{2, 4, 8\}}$ (Small)* | *0.09M* | *63.28* | *82.09* | *69.71* | *71.69* |
> | **$\text{TTL}_{\text{large}}$** (Rank=42) | 0.38M | 58.51 | 77.60 | 68.52 | 68.21 |
> | **$\text{TTL}_{\text{default}}$** (Rank=16) | 0.15M | 60.51 | 77.54 | 69.20 | 69.08 |
> | **$\text{TTL}_{\text{small}}$** (Rank=10) | 0.09M | 59.01 | 77.69 | 67.96 | 68.22 |

---

> ### Author Response · Authors · 2025-11-27
> **Response to Reviewer AXZk's Thoughtful Questions (Part 2/2)**
>
> **2. Questions regarding the inference overhead analysis.**
>
> **Source of superior performance: Multi-Rank Design**
>
> As discussed in Q1, MuRA's advantage stems from dynamically tailoring adaptation ranks to match input complexity, thereby effectively unlocking the model's full potential.
>
> **Source of superior efficiency: Deep-Layer Design**
>
> MuRA achieves high efficiency by restricting adaptation to the deepest layer, minimizing the backpropagation path to significantly reduce computational and memory overhead.
>
> **Why MuRA Enables Deepest-Layer Adaptation While Single-Rank Methods Struggle**
>
> As demonstrated in the table below, MuRA and single rank methods (such as TTL) exhibit divergent behaviors regarding adaptation depth. While single-rank methods suffer from performance degradation when restricted to the deepest layer, MuRA achieves its peak accuracy (72.98%) in this specific setting.
>
> We hypothesize that deep layers in VLMs are highly sensitive to updates. The single rank methods (TTL) likely lack the flexibility to handle this sensitivity, leading to performance degradation. MuRA, however, uses a dynamic router to provide adaptive updates, allowing it to function effectively at the deepest layer. This enables MuRA to unlock the efficiency benefits of deep-layer adaptation that are inaccessible to single rank methods.
>
> | Method | Layers 1-4 | Layers 5-8 | Layers 9-10 | Layer 11 | Layer 12 |
> | :--- | :---: | :---: | :---: | :---: | :---: |
> | Single-Rank | **67.87** | 67.46 | 67.71 | 67.78 | 67.51 |
> | **MuRA (Ours)** | 60.08 | 56.76 | 70.56 | 70.77 | **72.98** |
>
> **Comparison with additional single-rank configurations**
>
> As suggested, we compared MuRA against $TTL_{mean}$ (matching MuRA's average rank of 15) and $TTL_{micro}$ (matching MuRA's minimum rank of 2).
>
> This demonstrates that MuRA achieves the best trade-off. While $TTL_{micro}$ represents the efficiency upper bound, MuRA closely approaches this limit with competitive throughput and memory, all while delivering a substantial performance gain (+4.21%).
>
> | Method | FLOPs | Memory | Throughput | Avg. Accuracy* |
> | :--- | :--- | :--- | :--- | :--- |
> | **$TTL_{mean}$** (Rank=15) | 2.25 TFLOPs | 2.42 GB | 6.25 samples/s | 68.63 |
> | **$TTL_{micro}$** (Rank=2) | 2.25 TFLOPs | **1.96 GB** | **13.32 samples/s** | 67.48 |
> | **MuRA (Ours)** (Dynamic) | 2.40 TFLOPs | 2.05 GB | 11.26 samples/s | **71.69** |
>
> *Note: Avg. Accuracy is computed across ImageNet-A, ImageNet-R, and UCF101.*
>
>
>
> We hope these additional results fully address the reviewer's questions regarding the mechanisms behind MuRA's performance and efficiency.
>
> **3. Conclusion**
>
> In summary, MuRA addresses the limitations of static approaches by dynamically matching adaptive capacity (rank) to varying input complexities. Furthermore, our specialized designs—specifically Multi-Rank Adaptation, Deepest-Layer Adaptation, Multi-Rank Orthogonal Decomposition, and Continuous Router Updating—underpin MuRA’s comprehensive superiority in efficiency, robustness, and performance:
>
> * **Multi-Rank Adaptation:** We introduce an MoE framework featuring diverse rank combinations to adaptively allocate adapters with adequate capacity for each input complexity, achieving superior performance.
> * **Deepest-Layer Adaptation:** By restricting adaptation exclusively to the deepest visual layer, MuRA minimize the gradient backpropagation path to achieve the best efficiency.
> * **Multi-Rank Orthogonal Decomposition:** To enrich the pre-trained knowledge for different experts and stabilize the TTA optimization process, we adopt Multi-Rank Orthogonal Decomposition for initialization.
> * **Continuous Router Updating:** To exploit sample-shared knowledge in the TTA scenario, we introduced the Continuous Router Updating mechanism to continuously accumulate parameter updates across input samples, acquiring a robust "Complexity-Capacity Mapping".
>
> Given this demonstrated value and novelty, we respectfully invite the reviewer to reconsider the rating.

---

### Author Response · Authors · 2025-12-02
**Rebuttal Summary**

Dear Area Chair,

Given the recent reversion of review scores, we understand the increased workload during this period. To assist your final assessment, we provide a summary of our **technical novelty**, the **reviewer consensus on strengths**, our **key revisions**, and the **updated reviewer assessments**.

### 1. Technical Novelty
We are the first to identify that dynamic rank adaptation is pivotal for Test-Time Adaptation (TTA), as different inputs require varying degrees of adaptation capacity (rank). To address this, we propose MuRA, which demonstrates comprehensive superiority in efficiency, robustness, and accuracy through four specialized mechanisms:

* **Multi-Rank Adaptation:** We introduce a MoE framework featuring diverse rank alignment to adaptively allocate adequate capacity for differing input complexities. As shown in **Fig. 6a**, this MuRA outperforms homogeneous (same-rank) settings, and **Tab. 3 & 10** confirms it significantly outperforms single-rank methods even with comparable parameter budgets.
* **Deepest-Layer Adaptation:** To meet the stringent efficiency requirements of TTA, we strategically restrict adaptation to the deepest visual layer. **Fig. 1** demonstrate that this targeted design minimizes the gradient backpropagation path, achieving the optimal balance between effectiveness and efficiency.
* **Continuous Router Updating (CRU):** To exploit sample-shared knowledge, we introduce the CRU mechanism to continuously accumulate parameter updates across samples, acquiring a robust "Complexity-Capacity Mapping". The efficacy of this strategy is validated in **Tab. 3** and **Fig. 8, 10, & 12**.
* **Multi-Rank Orthogonal Decomposition (MROD):** To enrich pre-trained knowledge and stabilize the TTA optimization process, we utilize MROD initialization. Its superiority over random or standard initialization is evidenced in **Fig. 5** and **Tab. 3**.

### 2. Reviewer Consensus on Strengths
We sincerely appreciate the positive assessments from all reviewers:

* **Reviewer AXZk** commended the "comprehensive experiments" and "clarity of technical depth".
* **Reviewer 2MGn** found the paper "well-written and easy to understand," describing MuRA as an "interesting form of TTA" with "competitive performance".
* **Reviewer iFPH** highlighted our "practical, lightweight design" and "competitive results on multiple benchmarks," noting that the "framing of rank sensitivity" is clear and the motivation is "intuitive".
* **Reviewer nRHo** praised the work as a "well-designed, cohesive method" with "clear diagnosis & evidence".

### 3. Key Revisions
We strictly followed the reviewers' suggestions to improve the paper's quality and have incorporated substantial updates into the revised PDF. Major updates include:

* **Source of Superiority [AXZk]:** We demonstrated that MuRA's superiority stems from dynamically aligning ranks with input complexity rather than increasing the number of adapters or parameters, evidenced by comparisons against 'multiple same-rank adapters' (**Fig. 6a**) and parameter-matched baselines (**Tab. 10**).
* **Source of Efficiency [AXZk]:** We demonstrated that single-rank methods degrade at the deepest layer (**Tab. 11**). MuRA mitigates this, uniquely unlocking the efficiency benefits of deep-layer adaptation.
* **Robustness with Limited Views [iFPH]:** We added evaluations under limited augmentation views (**Fig. 11**). MuRA maintains SOTA accuracy and efficiency even in extreme settings.
* **MROD Advantages [iFPH]:** We provided controlled comparisons against Random and PCA-based construction, validating the critical design of SVD-based construction (MROD) (**Tab. 12**).
* **CRU Advantages [iFPH, nRHo, 2MGn]:** Sensitivity analysis (**Fig. 10**) and Continuous Learning evaluations (**Fig. 12**) confirm CRU's superiority. We further visualized rank routing evolution (**Fig. 13 & 14**) for interpretability and provided detailed pseudo-code (**Algorithm 1**).
* **Comparison with Latest Methods [2MGn]:** We compared MuRA against the most recent methods, confirming MuRA remains SOTA (**Tab. 9**).
* **Presentation [nRHo]:** We have improved figure quality and layout.

### 4. Updated Reviewer Assessments
Following the rebuttal phase, the status of the reviewers is as follows:

* **Reviewer AXZk:** Acknowledged most concerns were addressed and raised positive follow-up inquiries. We fully clarified these with additional experiments; however, due to the conclusion of the rebuttal period, the reviewer could not provide a further feedback.
* **Reviewer 2MGn:** Confirmed that all questions were addressed and maintained their positive Accept rating.
* **Reviewers iFPH & nRHo:** Regrettably, due to the abrupt suspension of the rebuttal process, these reviewers did not participate in further discussion.

We respectfully request the AC to consider **MuRA's core value**, our **key revisions**, and the **available reviewers' feedback** when drafting the meta-review.

Best regards,
Authors

---

### Meta-Review · Area_Chair_crRS · 2025-12-31

**Summary:**

The reviewers' concerns on the practical utility and generalizability of the MuRA framework are not addressed well. A critical, unresolved concern raised by Reviewer iFPH is the method's heavy reliance on a 64-view augmentation strategy (63 augmentations per image) to achieve its performance gains. The reviewer noted that this creates a massive computational bottleneck for real-time Test-Time Adaptation (TTA), and the authors failed to provide a convincing wall-clock time analysis or proof that the method remains competitive in low-latency settings with fewer views. Additionally, Reviewer 2MGn pointed out a significant gap in architectural generalizability; the method was only validated on CLIP-style Vision Transformers (ViT). The authors' claim that extending the approach to ResNets or LLMs is straightforward remained speculative, as no empirical evidence was provided to prove the framework's effectiveness across different backbones.

**Reviewer Concerns:**

The authors were successful in clarifying the following technical and procedural issues: rank diversity vs. capacity (Reviewer AXZk), initialization effectiveness (Reviewer iFPH), CL-style robustness (Reviewer nRHo), and algorithmic transparency (Reviewer iFPH).

Despite the thorough rebuttal, two major points remained unresolved or insufficiently clarified: real-world efficiency (Reviewer iFPH) and cross-architecture generalizability (Reviewer 2MGn).

**Reviewer Scores:**

Reviewer AXZk: This reviewer would raise his/her rating to 6.

Reviewer 2MGn: This reviewer confirmed to keep the current rating 6.

Reviewer iFPH: The reviewer is likely to keep the current rating 4 because the efficiency issue was not addressed.

Reviewer nRHo: The reviewer is likely to keep the current rating 4 because the addition of Figure 12 may not be fully satisfactory (showing the result for only one sequence).

---

### Decision · Program_Chairs · 2026-01-26

Reject